# β-catenin signaling modulates the tempo of dendritic growth of adult-born hippocampal neurons

Jana Heppt[1] (iD), Marie-Theres Wittmann[1,2] (iD), Iris Schäffner[1] (iD), Charlotte Billmann[1], Jingzhong Zhang[3,4], Daniela Vogt-Weisenhorn[3], Nilima Prakash[3,5], Wolfgang Wurst[3], Makoto Mark Taketo[6] & Dieter Chichung Lie[1,*] (iD)

## Abstract

In adult hippocampal neurogenesis, stem/progenitor cells generate dentate granule neurons that contribute to hippocampal plasticity. The establishment of a morphologically defined dendritic arbor is central to the functional integration of adult-born neurons. We investigated the role of canonical Wnt/β-catenin signaling in dendritogenesis of adult-born neurons. We show that canonical Wnt signaling follows a biphasic pattern, with high activity in stem/progenitor cells, attenuation in immature neurons, and reactivation during maturation, and demonstrate that this activity pattern is required for proper dendrite development. Increasing β-catenin signaling in maturing neurons of young adult mice transiently accelerated dendritic growth, but eventually produced dendritic defects and excessive spine numbers. In middle-aged mice, in which protracted dendrite and spine development were paralleled by lower canonical Wnt signaling activity, enhancement of β-catenin signaling restored dendritic growth and spine formation to levels observed in young adult animals. Our data indicate that precise timing and strength of β-catenin signaling are essential for the correct functional integration of adult-born neurons and suggest Wnt/β-catenin signaling as a pathway to ameliorate deficits in adult neurogenesis during aging.

**Keywords** adult neurogenesis; aging; dendrite; hippocampus; Wnt signaling
**Subject Categories** Neuroscience; Signal Transduction
**The EMBO Journal (2020) 39: e104472**

## Introduction

In the adult mammalian hippocampus, neural stem/progenitor cells in the subgranular zone of the dentate gyrus (DG) undergo a complex sequence of proliferation, differentiation, and maturation steps to add dentate granule neurons to the hippocampal network. A key step for the functional integration of adult-born dentate granule neurons is the development of a morphologically highly stereotypic dendritic arbor to receive afferents in the molecular layer (Goncalves *et al*, 2016b). Disruption of dendritic arbor development of adult-born neurons is thought to contribute to cognitive and emotional deficits in aging, neurodegenerative, and neuropsychiatric diseases and to the development of an aberrant circuitry in epilepsy (Li *et al*, 2009; Sun *et al*, 2009; Winner *et al*, 2011; Kim *et al*, 2012; Murphy *et al*, 2012; Fitzsimons *et al*, 2013; Cho *et al*, 2015; Jessberger & Parent, 2015; Llorens-Martin *et al*, 2015; Trinchero *et al*, 2017; Kerloch *et al*, 2018).

In young adult mice, the dendritic arbor of adult-born dentate granule neurons is largely established within the first 3–4 weeks of development. Around 10 days after their birth, new neurons feature a basic dentate granule neuron architecture with an apical dendrite spanning the dentate granule cell layer and initial branching in the inner molecular layer. Dendritic growth with further branching and dendritic extension into the outer molecular layer is maximal during the first 2–3 weeks and is followed by a period of pruning of excessive dendritic branches to attain the highly stereotypic dentate granule neuron morphology (Zhao *et al*, 2006; Kleine Borgmann *et al*, 2013; Sun *et al*, 2013; Goncalves *et al*, 2016a). Several factors including hippocampal network activity, transcription factors, cytoskeletal regulators, neurotransmitters, and signaling molecules were found to modulate dendrite morphology of adult-born neurons (Ge *et al*, 2006; Bergami *et al*, 2008; Gao *et al*, 2009; Jagasia *et al*, 2009; Ma *et al*, 2009; Piatti *et al*, 2011; Llorens-Martin *et al*, 2013;

1 Institute of Biochemistry, Emil Fischer Center, Friedrich-Alexander Universität Erlangen-Nürnberg, Erlangen, Germany
2 Institute of Human Genetics, Universitätsklinikum Erlangen, Friedrich-Alexander-Universität Erlangen-Nürnberg, Erlangen, Germany
3 Institute of Developmental Genetics, Helmholtz Center Munich, German Research Center for Environmental Health, Neuherberg, Germany
4 Suzhou Institute of Biomedical Engineering and Technology (SIBET), Chinese Academy of Sciences, Suzhou, China
5 Hamm-Lippstadt University of Applied Sciences, Hamm, Germany
6 Division of Experimental Therapeutics, Graduate School of Medicine, Kyoto University, Kyoto, Japan
*Corresponding author. Tel: +49 9131 85 24622; E-mail: chi.lie@fau.de

Vadodaria *et al*, 2013; He *et al*, 2014; Trinchero *et al*, 2017). A complete understanding of the central pathways controlling dendrite development in adult hippocampal neurogenesis is, however, still missing.

Wnt proteins are key regulators of adult hippocampal neurogenesis (Lie *et al*, 2005; Qu *et al*, 2010, 2013; Jang *et al*, 2013; Seib *et al*, 2013; Arredondo *et al*, 2019). Current data indicate that Wnts control different stages of adult neurogenesis via distinct pathways: While early developmental steps such as proliferation and fate determination of precursors are regulated by canonical Wnt/β-catenin signaling (Lie *et al*, 2005; Kuwabara *et al*, 2009; Karalay *et al*, 2011; Qu *et al*, 2013), late developmental steps such as neuronal maturation and morphogenesis are thought to be primarily regulated by non-canonical Wnt signaling pathways (Schafer *et al*, 2015; Arredondo *et al*, 2019). Supporting a sequential action of distinct Wnt pathways is the finding that early adult-born neuron development and the initiation of neuronal morphogenesis are accompanied by the attenuation of canonical Wnt/β-catenin signaling activity and the increased activity of the non-canonical Wnt/planar cell polarity (PCP) signaling pathway (Schafer *et al*, 2015). However, the observations i) that pathologies associated with aberrant Wnt/β-catenin activity are paralleled by dendritic growth defects of adult-born dentate granule neurons (Duan *et al*, 2007; Singh *et al*, 2011; Murphy *et al*, 2012; Llorens-Martin *et al*, 2013; De Ferrari *et al*, 2014; Qu *et al*, 2017; Martin *et al*, 2018) and ii) that ablation of β-catenin from developing dentate granule neurons in juvenile mice causes massive dendritic defects and neuronal death (Gao *et al*, 2007), raise the possibility that Wnt/β-catenin signaling fulfills important functions during late steps of adult hippocampal neurogenesis.

We here report that attenuation of canonical Wnt signaling in early immature neurons is followed by reactivation of the pathway during maturation, resulting in a biphasic pattern of canonical Wnt signaling activity in the adult neurogenic lineage. We also show that this biphasic activity pattern is essential to ensure correct dendrite development and that β-catenin signaling in maturing neurons modulates the tempo of dendritic growth and spine formation.

Finally, we demonstrate that countering the age-associated decrease in canonical Wnt/β-catenin signaling reverses dendritic growth and spine formation deficits of adult-born neurons in middle-aged mice. Thus, our data reveal a new function of β-catenin signaling in maturation of adult-born neurons and suggest canonical Wnt/β-catenin signaling as a candidate pathway to counteract age-related deficits in hippocampal neurogenesis-dependent plasticity.

# Results

### Canonical Wnt signaling exhibits biphasic activity during adult hippocampal neurogenesis

Analyses of different reporter mouse lines consistently revealed high activity of canonical Wnt signaling in the adult DG (O'Brien *et al*, 2004; Lie *et al*, 2005; Garbe & Ring, 2012) but were inconclusive regarding canonical Wnt signaling activity during different stages of adult-born neuron development (Garbe & Ring, 2012). To shed light on the activity pattern of canonical Wnt signaling in adult neurogenesis, we analyzed two different reporter mouse lines: BATGAL mice harbor the LacZ reporter gene downstream of seven TCF/LEF-binding sites and the minimal promoter-TATA box of the *siamois* gene (Maretto *et al*, 2003); Axin2$^{LacZ/+}$ mice heterozygously harbor the LacZ reporter in the endogenous locus of the bona fide canonical Wnt signaling target Axin2 (Lustig *et al*, 2002) (Fig 1A). We first analyzed the percentage of reporter-positive cells as a proxy for canonical Wnt signaling activity. Both reporter lines showed a qualitatively comparable pattern of canonical Wnt signaling activity (Fig 1B and C). In both lines, around one-fourth of the Nestin$^+$ radial glia-like stem/progenitor cells were reporter positive. Immature neurons that were identified by the expression of DCX rarely displayed reporter gene expression, whereas a high percentage of mature Calbindin$^+$ dentate granule neurons was positive for the reporter. The two reporter lines differed with regard to reporter activity in Tbr2$^+$ precursor cells. In BATGAL animals, the fraction of reporter-positive Tbr2$^+$ cells and Nestin$^+$ cells were comparable,

---

**Figure 1. Canonical Wnt signaling activity in the adult hippocampus.**

A   Schematic representation of the *LacZ* alleles of the BATGAL and Axin2$^{LacZ/+}$ reporter mice for canonical Wnt signaling activity.

B   Representative images showing co-expression of the stage-specific markers Nestin, Tbr2, Doublecortin (DCX), and Calbindin (all in green), with the β-galactosidase (β-Gal, red) reporter in 8-week-old BATGAL and Axin2$^{LacZ/+}$ mice. Nuclei are counterstained with DAPI (in blue). Scale bar = 10 μm. Insets show a 1.5× magnification of selected cells (position indicated by dashed box). Scale bar = 5 μm.

C   Fraction of Nestin-, Tbr2-, DCX- and Calbindin-positive cells expressing β-Gal in BATGAL and Axin2$^{LacZ/+}$ animals show stage-specific canonical Wnt signaling during adult hippocampal neurogenesis (BATGAL: Tbr2 vs. DCX $P = 0.0056$, DCX vs. Calbindin $P < 0.0001$; Axin2$^{LacZ}$: Nestin vs. Tbr2 $P = 0.0003$, Tbr2 vs. DCX $P = 0.0115$, DCX vs. Calbindin $P < 0.0001$; $n = 3$ animals per mouse model and marker).

D   Detection of the corrected total cell fluorescence (CTCF) of the β-Gal reporter in Nestin-, Tbr2-, DCX-, and Calbindin-positive cells in BATGAL mice corroborate stage-specific activity pattern of canonical Wnt signaling during lineage progression (Nestin vs. Tbr2 $P < 0.0001$, Tbr2 vs. DCX $P = 0.0029$, DCX vs. Calbindin $P < 0.0001$; Nestin: $n = 236$ cells, Tbr2: $n = 134$ cells, DCX: $n = 200$ cells, Calbindin: $n = 300$ cells).

E   BrdU pulse chase scheme. BrdU was injected intraperitoneally (i.p.) three times: (i) every 2 h for 30-min time point (0 days post-injection [dpi]) (ii) every 24 h for all other time points. For the 30-min time point animals were sacrificed 30 min after the final BrdU injection.

F   Quantification of BrdU$^+$ cells expressing β-Gal reveals biphasic activity of canonical Wnt signaling during adult hippocampal neurogenesis (0 vs. 7 dpi $P = 0.0036$, 7 vs. 28 dpi $P = 0.0115$, 7 vs. 42 dpi $P = 0.0007$; 0 dpi time point; $n = 4$ animals, 3 dpi $n = 3$ animals, 7 dpi $n = 3$ animals, 14 dpi $n = 4$ animals, 28 dpi $n = 6$ animals, and 42 dpi $n = 6$ animals).

G   Total cell fluorescence of β-Gal in BrdU$^+$ cells indicate a biphasic pattern in canonical Wnt signaling strength during lineage progression (0 vs. 3 dpi $P < 0.0001$, 7 vs. 14 dpi $P < 0.0001$, 14 vs. 28 dpi $P < 0.0001$, 28 vs. 42 dpi $P = 0.0333$; 0 dpi $n = 194$ cells, 3 dpi $n = 245$ cells, 7 dpi $n = 224$ cells, 14 dpi $n = 72$ cells, 28 dpi $n = 103$ cells, and 42 dpi $n = 114$ cells).

Data information: Data represented as mean ± SEM. Significance was determined using Kruskal–Wallis test followed by Dunn's multiple comparisons test, and significance levels are displayed in GP style (*$P < 0.0332$, **$P < 0.0021$, and ****$P < 0.0001$).

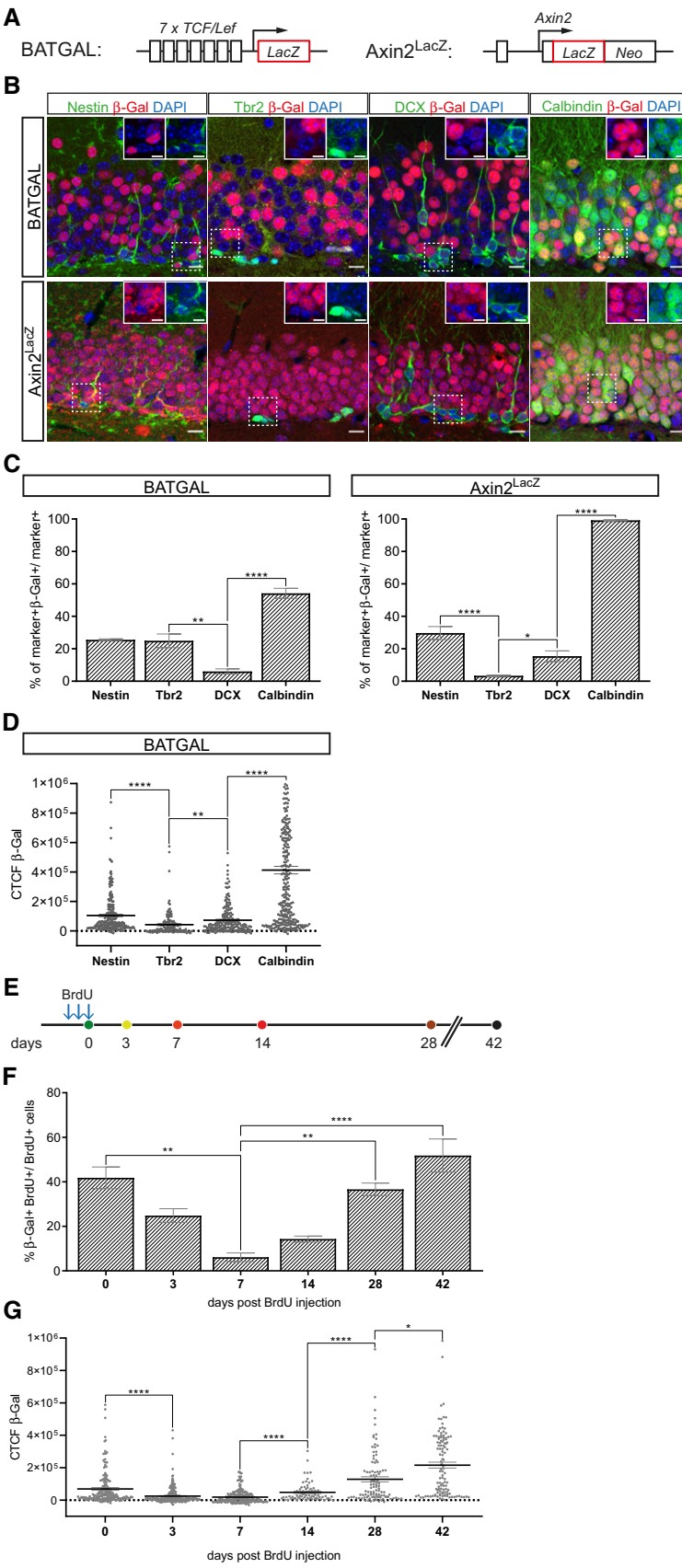

**Figure 1.**

suggesting that canonical Wnt signaling activity was sustained during lineage progression of radial glia-like stem/progenitor cells toward fast-proliferating precursor cells. In contrast, the reporter pattern in Axin2$^{LacZ/+}$ mice suggested that canonical Wnt signaling activity was readily attenuated in Tbr2$^+$ cells (Fig 1B and C). We further determined signal intensity of the reporter in BATGAL mice as an additional measure of canonical Wnt signaling activity (Fig 1D). Tbr2$^+$ cells had on average lower reporter expression than Nestin$^+$ cells, which supports the notion that attenuation of canonical Wnt signaling activity is initiated at the level of Tbr2$^+$ precursor cells. Calbindin$^+$ cells showed on average the highest level of reporter signal, strongly indicating reactivation of canonical Wnt signaling in mature neurons.

To further assess the time course of canonical Wnt signaling activity, newborn cells in 8-week-old reporter mice were birth-dated with Bromodeoxyuridine (BrdU, Fig 1E). For this analysis, we focused on the BATGAL reporter strain, because Axin2 functions in a feedback loop as a negative regulator of canonical Wnt signaling (Lustig *et al*, 2002); consequently, heterozygous loss of Axin2 may affect the physiological time line of adult-born neuron development. BATGAL reporter activity was analyzed at time points that correspond approximately to the proliferating progenitor cell stage (30 min post-injection), the neuroblast stage (3 days post-injection [dpi]), early and mid-immature neuron stage (7 and 14 dpi, respectively), and the early and late mature neuron stage (28 and 42 dpi, respectively) (Jagasia *et al*, 2009; Snyder *et al*, 2009). Thirty minutes after BrdU injection, 42% of BrdU$^+$ cells showed reporter activity. This fraction dropped to 25 and 7% at the 3 and 7 dpi time points, respectively, to subsequently increase to 14% at 14 dpi, 38% at 28 dpi, and 52% at 42 dpi (Fig 1F). Furthermore, analysis of β-galactosidase signal intensities in BrdU$^+$ cells showed that the average reporter expression was strongly increased at 28 and 42 dpi (Fig 1G). Collectively, these data indicate that canonical Wnt signaling activity in the adult neurogenic lineage is attenuated in Tbr2$^+$ precursor cells and immature DCX$^+$ neurons during the first week of development and is up-regulated around the second week during the maturation of DCX$^+$ neurons into Calbindin$^+$ neurons.

## Correct dendrite development of adult-born neurons is dependent on the timing and dosage of β-catenin signaling activity

We first asked whether the reactivation of canonical Wnt/β-catenin signaling during maturation was required for adult-born neuron development. We focused in particular on dendrite development given that reactivation of Wnt/β-catenin signaling activity overlapped with the period of maximal dendritic growth speed (Sun *et al*, 2013). In canonical Wnt signaling, stabilized β-catenin activates transcription through members of the TCF/LEF transcription factor family (Moon, 2004). To inhibit β-catenin-induced transcription, we transduced fast-proliferating precursor cells in the DG of young adult mice with the CAG-dnLEF-IRES-GFP Moloney Murine Leukemia retrovirus (MMLV), which bi-cistronically encodes for a dominant-negative LEF-mutant protein (dnLEF) and GFP (Karalay *et al*, 2011). To validate the ability of dnLEF to inhibit β-catenin-induced transcription in adult-born neurons, the CAG-dnLEF-IRES-GFP MMLV was injected into the DG of BATGAL reporter mice. BATGAL mice injected with an MMLV encoding for GFP (CAG-GFP) served as control (Fig EV1A). Comparison of the reporter signal on day 17 post-viral injection between dnLEF-transduced cells and control transduced cells demonstrated that expression of dnLEF inhibited canonical Wnt signaling-induced transcriptional activity (Fig EV1B and C).

We next co-injected young adult (i.e., 8-week-old) mice with CAG-dnLEF-IRES-GFP and a MMLV encoding for RFP (CAG-RFP). Animals were analyzed 17 dpi (Fig 2A). Compared to RFP$^+$ GFP$^-$ control neurons, CAG-dnLEF-IRES-GFP-transduced neurons displayed a more immature morphology, with shorter total dendritic length likely caused by shorter terminal dendrites as indicated by the analysis of branch point number and Sholl analysis. In addition, dnLEF-expressing neurons almost invariably exhibited basal dendrites, a transient feature of developing dentate granule neurons and morphological indicator of immaturity (Ribak *et al*, 2004) (Fig 2B–E). Hence, dnLEF-expressing neurons lagged behind control neurons with regard to dendrite development, indicating that canonical β-catenin-dependent signaling is required for the timely execution of the dendrite development program.

**Figure 2. Loss of canonical Wnt signaling and sustained canonical Wnt signaling impair dendritogenesis of adult-born neurons.**

A   Experimental scheme of retroviral injection paradigm. Adult mice were stereotactically co-injected with the MMLV CAG-dnLEF-IRES-GFP (dnLEF) and CAG-RFP (control) and were sacrificed 17 dpi.

B   Representative images of transduced adult-born neurons at 17 dpi. CAG-dnLEF-IRES-GFP (dnLEF, green) and CAG-RFP (control, red) double transduced cells (arrows) and CAG-RFP single transduced cells (arrowheads). Scale bar = 20 μm.

C   Representative reconstructions of control and dnLEF neurons. Scale bar = 20 μm.

D   Analysis of morphology showed a reduction in dendritic length in dnLEF-transduced neurons (*P* < 0.0001), while the number of branch points remained comparable (*P* = 0.7914). Sholl analysis displayed a reduction in dendritic complexity in dnLEF-transduced neurons and indicated decreased growth of terminal dendritic branches (*P* < 0.0001; control: *n* = 20 cells from three animals, dnLEF: *n* = 38 cells from six animals).

E   dnLEF neurons displayed basal dendrites (*P* < 0.0001; control: *n* = 20 cells from three animals, dnLEF: *n* = 38 cells from six animals).

F   Schematic representation of the conditional alleles of the Ctnnb1$^{(ex3)fl}$ (β-cat$^{ex3}$) mouse model and the retroviral paradigm used to analyze canonical Wnt signaling gain of function in neural progenitors. Control animals harbor the wild-type allele for Ctnnb1.

G   Representative images depicting CAG-GFP-IRES-Cre transduced adult-born neurons in control and β-cat$^{ex3}$ mice at 17 dpi. Scale bar = 20 μm.

H   Representative reconstructions of control and β-cat$^{ex3}$ neurons. Scale bar = 20 μm.

I   Quantification showed decreased dendritic length of β-cat$^{ex3}$ neurons (*P* < 0.0001); no difference was apparent in number of branch points (*P* = 0.3256). Sholl analysis displayed a less complex dendritic tree of β-cat$^{ex3}$ neurons (*P* < 0.0001; control: *n* = 20 cells from five animals, β-cat$^{ex3}$: *n* = 20 cells from five animals).

J   The number of basal dendrites was increased in β-cat$^{ex3}$ neurons (*P* = 0.0003; control: *n* = 20 cells from five animals, β-cat$^{ex3}$: *n* = 20 cells from five animals).

Data information: Data represented as mean ± SEM, significance was determined using two-way ANOVA for Sholl analysis and two-tailed Mann–Whitney *U*-test for all other analyses, and significance levels were displayed in GP style (***$P$ < 0.0002 and ****$P$ < 0.0001).

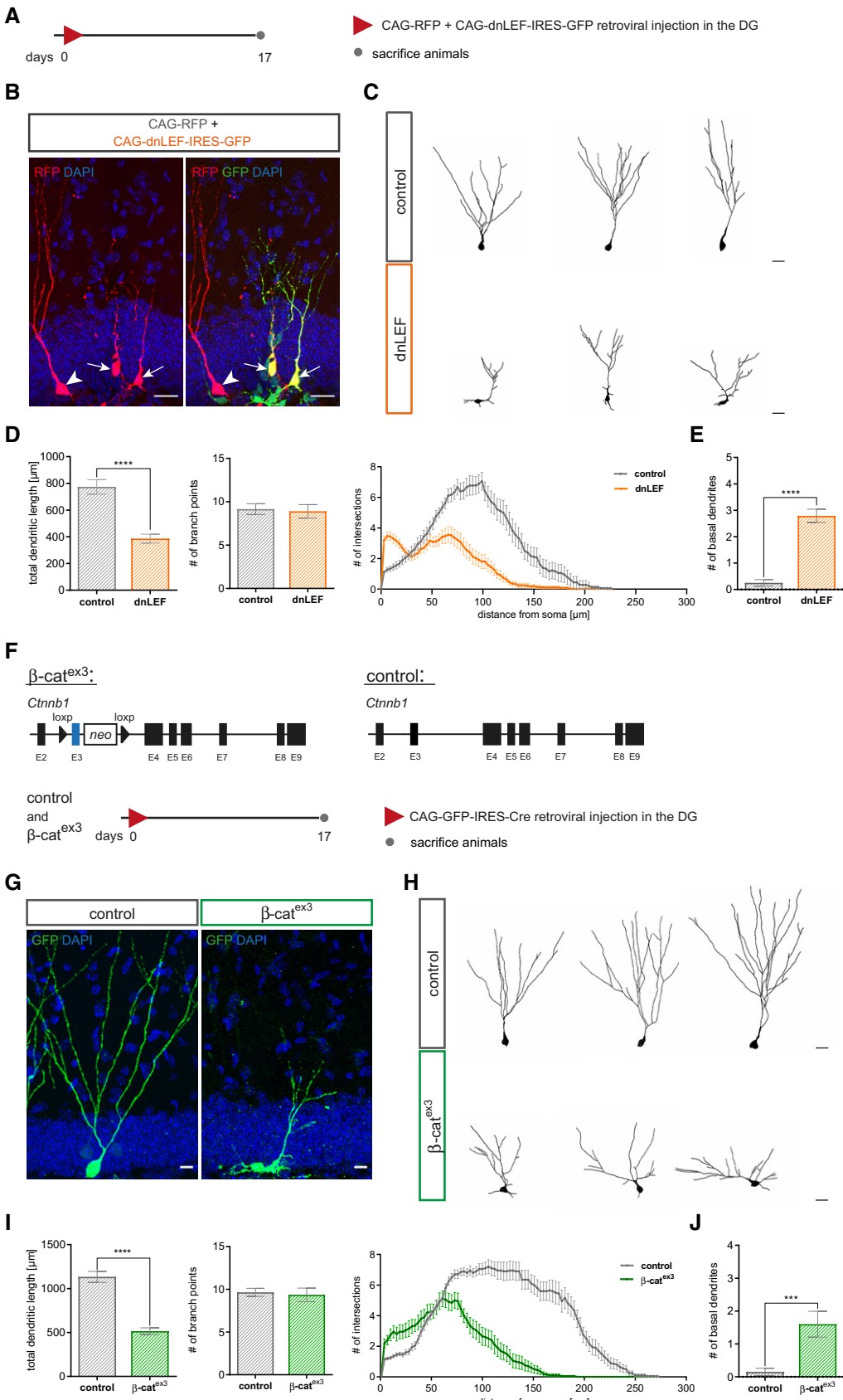

**Figure 2.**

Dendritic arborization in the molecular layer provides the structural basis for formation of glutamatergic synaptic input from the entorhinal cortex. Previous studies identified glutamatergic input as a critical signal for survival of adult-born neurons (Tashiro *et al*, 2006a). To determine the long-term survival of dnLEF-transduced neurons, mice were co-injected with CAG-dnLEF-IRES-GFP and CAG-RFP and analyzed at 17 and 42 dpi (Fig EV2A). At 42 dpi, the number of dnLEF-transduced neurons was dramatically reduced and the ratio of GFP$^+$ to RFP$^+$ cells dropped from approximately 1.5 at 17 dpi to 0.5 at 42 dpi, indicating that dnLEF expression strongly decreased survival of adult-born neurons. Moreover, dnLEF-expressing neurons featured a dendritic morphology with subtle alterations in the Sholl analysis (Fig EV2B–D).

Given that inhibition of canonical Wnt/β-catenin signaling retarded dendrite growth, we hypothesized that activation of β-catenin signaling promotes dendritic development. The Ctnnb1$^{(ex3)fl}$ mouse mutant (hereafter called β-cat$^{ex3}$) allows for Cre-recombinase-induced expression of a stabilized form of β-catenin (Harada *et al*, 1999). To enhance β-catenin-dependent signaling, we induced recombination in fast-dividing precursors by stereotactic injection of a MMLV bi-cistronically encoding for GFP and Cre-recombinase (CAG-GFP-IRES-Cre). Ctnnb1$^{(ex3)wt}$ mice injected with the CAG-GFP-IRES-Cre MMLV served as controls. Animals were analyzed 17 days post-viral injection (Fig 2F). Surprisingly, transduced neurons (GFP$^+$; DCX$^+$) in β-cat$^{ex3}$ mice featured an immature dendritic arbor that was characterized by a decrease in total dendritic length, irregular short neurites, and a reduced complexity in the Sholl analysis (Fig 2G–J), demonstrating that failure to attenuate β-catenin signaling in fast-dividing progenitor cells disrupted physiological dendritogenesis.

To determine how failure to attenuate β-catenin signaling affected the long-term fate of neurons, a second cohort of mice was analyzed at 42 days post-viral injection (Fig EV2E). The number of transduced neurons was dramatically reduced in β-cat$^{ex3}$ mice, suggesting that continuous β-catenin-dependent signaling impaired long-term survival of adult-born neurons. Moreover, Sholl analysis revealed subtle alterations in dendrite morphology of the remaining β-cat$^{ex3}$ neurons (Fig EV2F–H).

Recombination of the β-cat$^{ex3}$ locus in fast-dividing progenitor cells abolishes the early attenuation of canonical Wnt signaling in the adult neurogenic lineage (Fig 1 and (Schafer *et al*, 2015)). To allow for the initial attenuation of canonical Wnt signaling, we targeted recombination of the β-cat$^{ex3}$ locus to a later developmental time point. To this end, we generated the DCX::CreER$^{T2}$; CAG-CAT-GFP; β-cat$^{ex3}$ mouse line (hereafter called β-cat$^{ex3}$ iDCX) that allowed for tamoxifen-induced expression of stabilized β-catenin in DCX-expressing immature neurons and for tracing of recombined cells by expression of a GFP reporter (Harada *et al*, 1999; Nakamura *et al*, 2006; Zhang *et al*, 2010). DCX::CreER$^{T2}$; CAG-CAT-GFP; Ctnnb1$^{(ex3)wt}$ mice harboring wild-type alleles for Ctnnb1 served as controls (Fig EV3A). We first validated the tamoxifen-mediated induction of β-catenin-dependent transcription in β-cat$^{ex3}$ iDCX. To this end, the β-cat$^{ex3}$ iDCX and the control mouse line were crossed with the BATGAL reporter mouse line. Recombination was induced in 8-week-old mice by injection of tamoxifen on five consecutive days. Animals were analyzed 13 days after the tamoxifen pulse (Fig EV3B). GFP$^+$ recombined cells in β-cat$^{ex3}$ iDCX; BATGAL mice showed on average higher reporter expression levels than recombined cells in control mice, indicating that tamoxifen-induced recombination increased canonical Wnt signaling

activity in β-cat$^{ex3}$ iDCX mice (Fig EV3C and D). Numerous non-recombined cells in the granule cell layer, which were most likely mature dentate granule neurons, showed higher reporter expression than recombined neurons (Fig EV3C), suggesting that β-cat$^{ex3}$-driven β-catenin signaling activity did not exceed physiological Wnt/β-catenin signaling activity levels found in dentate granule neurons.

Next, young adult (i.e., 8-week-old) β-cat$^{ex3}$ iDCX and control mice were injected with tamoxifen for five consecutive days and analyzed 3 and 13 days later (Fig 3A). Recombined neurons were identified by co-expression of GFP$^+$ and the dentate granule neuron marker Prox1$^+$. Analysis at 3 days after recombination showed that enhanced β-catenin signaling activity stimulated the growth of terminal dendrite branches as evidenced by Sholl analysis, analysis of total dendritic length and number of branch points (Fig 3B and C). We then asked whether enhanced dendrite growth would produce an excessive dendrite arbor. To this end, animals were analyzed 13 days after recombination. At this time point, β-cat$^{ex3}$ iDCX neurons and control neurons showed a comparable branching pattern; surprisingly, β-cat$^{ex3}$ iDCX neurons showed a trend toward decreased total dendritic length, while Sholl analysis revealed a reduction in the length of the terminal dendrites (Fig 3D). These data suggest that increased β-catenin signaling in adult-born neurons of young adult mice transiently accelerated dendrite growth, but ultimately impeded full dendrite development.

To further evaluate the impact of enhanced β-catenin signaling on neuronal maturation, we determined dendritic spine density as a proxy of the development of the glutamatergic postsynaptic compartment as well as the expression of the immature neuron marker DCX and the mature neuron marker Calbindin. While dendritic spine density was comparable between β-cat$^{ex3}$ iDCX and control neurons at the 3-day time point (Fig 3E), β-cat$^{ex3}$ iDCX neurons bore a higher dendritic spine density at the 13-day time point (Fig 3E) suggesting that β-catenin signaling activity promoted spinogenesis. Three days after recombination 95% of recombined neurons (i.e., GFP$^+$ Prox1$^+$ cells) in control animals were DCX positive, while 5% expressed Calbindin. Activation of β-catenin signaling resulted only in a small difference in DCX and Calbindin expression: β-cat$^{ex3}$ iDCX animals showed a slightly decreased proportion of DCX$^+$ immature neurons (82%) and a mildly increased fraction of neurons expressing Calbindin (11%; Fig 3F and G). Thirteen days after recombination, expression of DCX and Calbindin was comparable between control and β-cat$^{ex3}$ iDCX mice (Fig 3G). These data indicate that enhanced β-catenin signaling activity in DCX$^+$ neurons of young adult mice was not sufficient to substantially accelerate the maturation-associated switch from DCX- to Calbindin- expression.

The DCX::CreER$^{T2}$ transgene targets β-catenin-dependent signaling activity to immature neurons of various birthdates. To more precisely understand the impact of enhanced β-catenin signaling on the developmental trajectory of adult-born neurons, we compared marker expression and dendrite parameters between neurons of a defined age. Neurons in β-cat$^{ex3}$ iDCX and control mice were birth-dated via BrdU pulse labeling or transduction with a MMLV encoding for RFP (CAG-RFP) prior to recombination. Animals received tamoxifen injections from day 10 until day 14 after birthdating. Consequently, BrdU- and RFP- labeled neurons were at most 14 days old at the time of recombination and thus were still in the phase of low or attenuated canonical Wnt signaling activity. Animals were analyzed three and 13 days after tamoxifen-induced

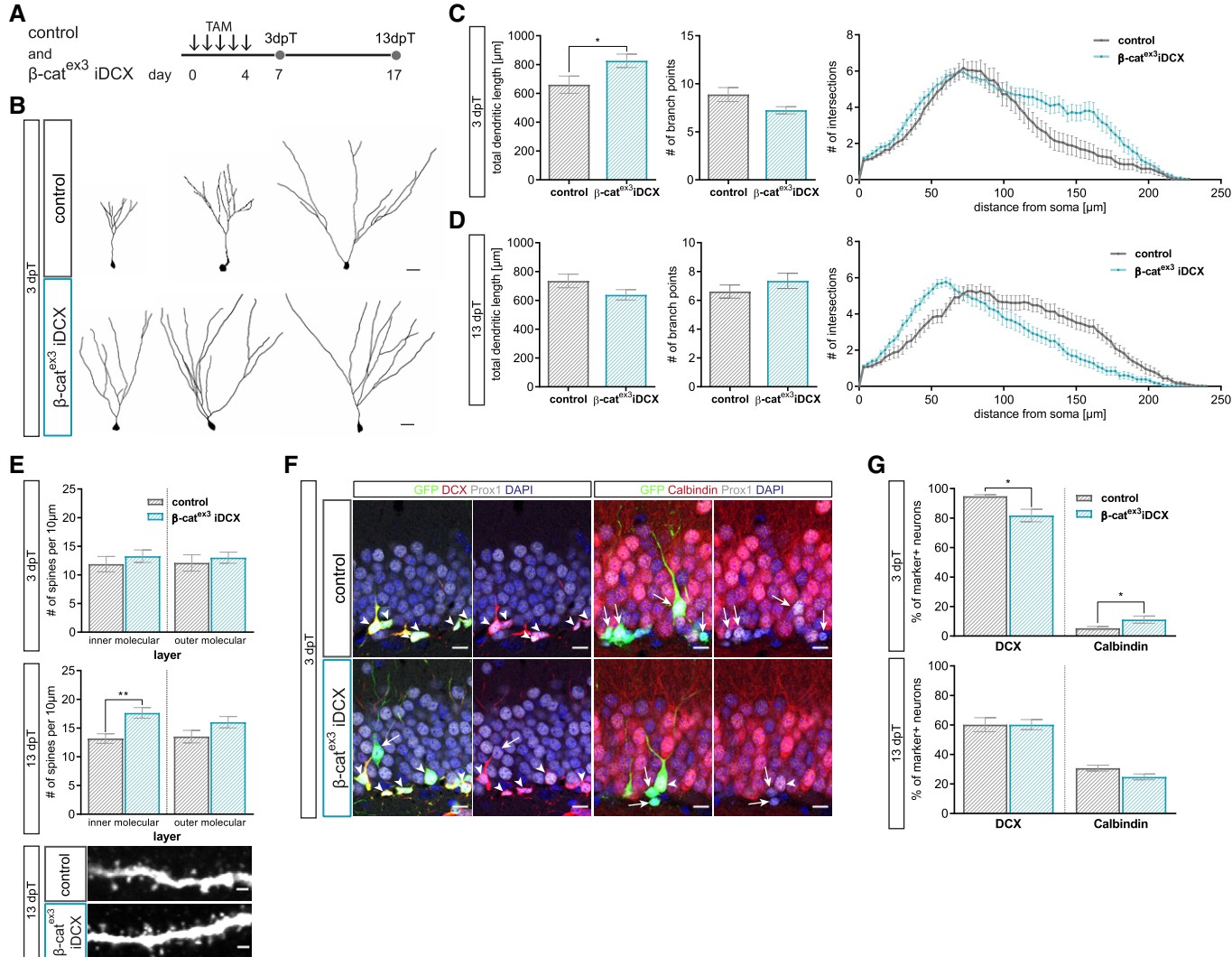

**Figure 3. Enhanced canonical Wnt signaling activity in immature neurons of young adult mice modulates dendritic growth and spine formation.**

A   Experimental paradigm: tamoxifen (TAM) was applied i.p. every 12 h for 5 days to control and β-cat$^{ex3}$ iDCX mice and animals were analyzed 3 and 13 days post-tamoxifen (dpT).

B   Representative reconstructions of control and β-cat$^{ex3}$ iDCX neurons at 3 dpT. Scale bar = 20 μm.

C   Analysis of morphology showed increased dendritic length in β-cat$^{ex3}$ iDCX neurons at 3 dpT ($P$ = 0.0320). The number of branch points was unaltered ($P$ = 0.0651). Sholl analysis indicated increased growth of terminal dendrite branches in neurons of β-cat$^{ex3}$ iDCX mice ($P$ < 0.0001; control: $n$ = 25 cells from nine animals; β-cat$^{ex3}$ iDCX: $n$ = 25 cells from eight animals).

D   Analysis of morphology showed no difference in dendritic length ($P$ = 0.1029) and branch point number ($P$ = 0.3726) between experimental groups at 13 dpT. Sholl analysis indicated a decrease in length of terminal dendrites in neurons of β-cat$^{ex3}$ iDCX mice ($P$ < 0.01; control: $n$ = 23 cells from nine animals, β-cat$^{ex3}$ iDCX: $n$ = 26 cells from eight animals).

E   Spine densities were comparable between experimental groups at 3 dpT (iML $P$ = 0.1512, oML $P$ = 0.3841). 13 dpT β-cat$^{ex3}$ iDCX neurons showed significantly increased spine density in the inner molecular layer (iML $P$ < 0.0001, oML $P$ = 0.0402; 3 dpT: control: $n$ = 20 cells from three animals, β-cat$^{ex3}$ iDCX: $n$ = 20 cells from four animals; 13 dpT control: $n$ = 20 cells from five animals, β-cat$^{ex3}$ iDCX: $n$ = 20 cells from seven animals). Representative images showing dendritic segments in the inner molecular layer of the DG at 13 dpT. Scale bars = 1 μm.

F   Representative images of recombined (GFP$^+$) neurons co-expressing Prox1 (gray) as a marker for neuronal fate and the stage-specific markers DCX (red) for immature neurons and Calbindin (red) for mature neurons. Arrows and arrowheads indicate marker-negative and marker-positive cells, respectively. Scale bar = 10 μm.

G   Fraction of DCX-expressing neurons is slightly reduced in β-cat$^{ex3}$ iDCX at 3 dpT ($P$ = 0.0338), while fraction of Calbindin expressing cells is slightly increased in β-cat$^{ex3}$ iDCX at 3 dpT ($P$ = 0.0345). At 13 dpT, no difference is visible in stage-specific marker expression (DCX $P$ = 0.8564, Calbindin $P$ = 0.0841; 3 dpT: control: $n$ = 12 animals, β-cat$^{ex3}$ iDCX: $n$ = 12 animals, 13 dpT: control: $n$ = 10 animals, β-cat$^{ex3}$ iDCX: $n$ = 14 animals).

Data information: Data represented as mean ± SEM, significance was determined using two-way ANOVA for Sholl analysis and two-tailed Mann–Whitney U-test for all other analyses, and significance levels were displayed in GP style (*$P$ < 0.0332 and **$P$ < 0.0021).

recombination (Fig 4A). Analysis of BrdU-labeled recombined neurons (GFP$^+$ BrdU$^+$) indicated that increased β-catenin activity did not affect the maturation-associated switch from DCX to Calbindin (Fig 4B). Three days after recombination, CAG-RFP-birthdated neurons with enhanced β-catenin activity (GFP$^+$ RFP$^+$) bore a more mature dendritic arbor with substantially increased total dendrite length and a higher number of branch points compared to control neurons (Fig 4C and D). As expected, control neurons showed an increase in dendritic length and complexity between the early and late time point. In contrast, neurons with enhanced β-catenin activity did not display signs of further dendrite growth and refinement but rather showed a reduction in dendritic complexity to a level that was significantly below the level of control neurons (Fig 4E and F). Thus, analysis of birthdated neurons confirmed that β-catenin signaling transiently accelerated dendrite growth but ultimately impeded dendrite development.

Collectively, our loss- and gain-of-function analyses demonstrate that physiological dendrite development of adult-born neurons

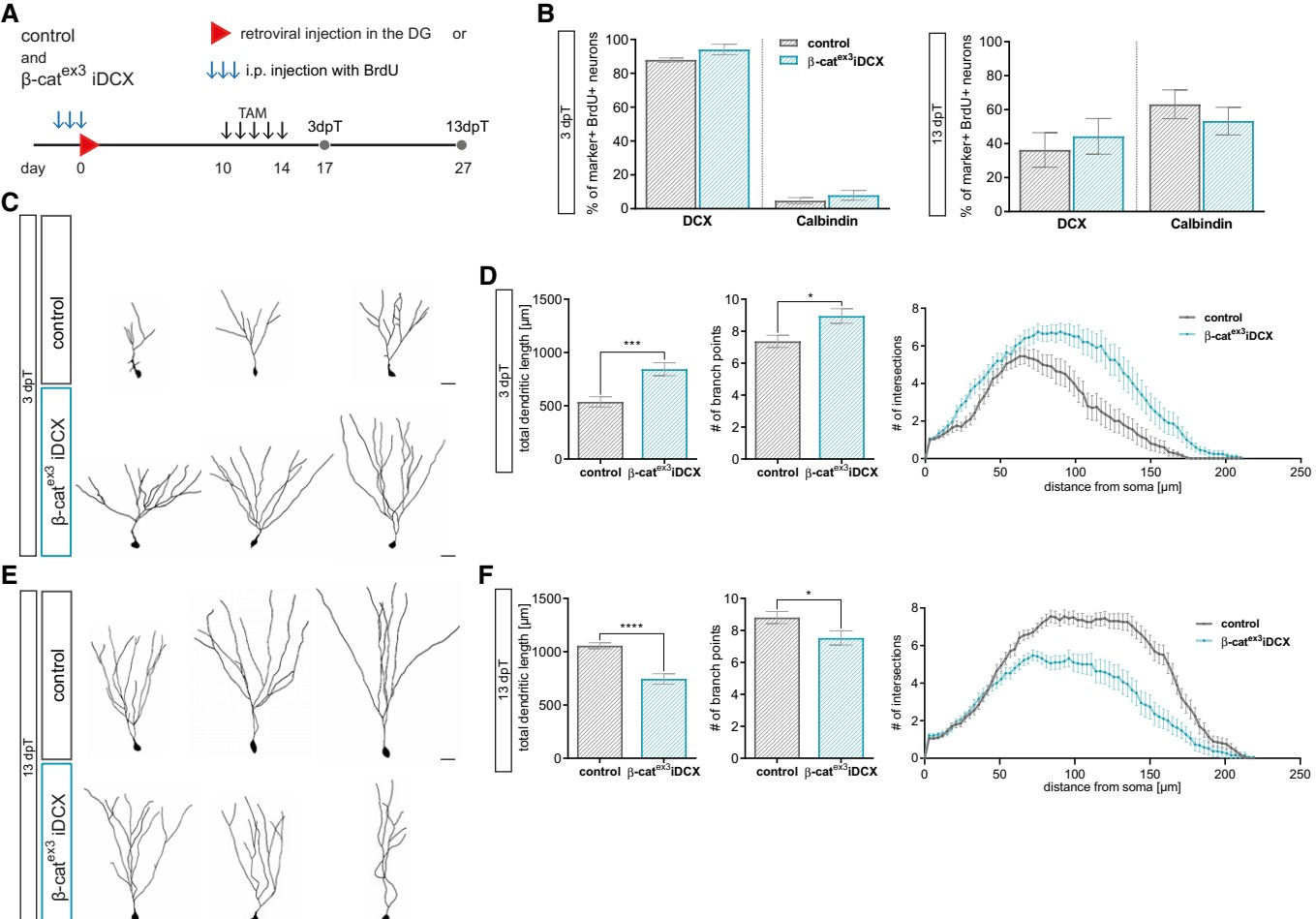

Figure 4. Analysis of birthdated immature neurons in young adult mice confirms modulation of dendritic growth and spine formation by canonical Wnt signaling activity.

A Experimental scheme for birthdating of adult-born neurons and recombination. For morphology analysis, CAG-RFP was stereotactically injected into the DG (C–F), for marker expression analysis, BrdU was injected i.p. every 24 h for 3 days (B). Tamoxifen was applied i.p. every 12 h for 5 days from day 10 to day 14. Mice were sacrificed 3 days post-tamoxifen (dpT) and 13 dpT, which corresponds to 17 and 27 dpi, respectively.

B Expression of DCX and Calbindin in BrdU$^+$ neurons was comparable between experimental groups at 3 and 13 dpT (control: $n$ = 5 animals, β-cat$^{ex3}$ iDCX: $n$ = 5 animals).

C Representative reconstructions of control and β-cat$^{ex3}$ iDCX neurons expressing RFP at 3 dpT. Scale bar = 20 μm.

D Quantification of dendritic length ($P$ = 0.0003), branch points ($P$ = 0.0433), and Sholl analysis ($P$ < 0.0001) showed a higher dendritic complexity of β-cat$^{ex3}$ iDCX neurons at 3 dpT (control: $n$ = 22 cells from four animals, β-cat$^{ex3}$ iDCX: $n$ = 19 cells from five animals).

E Representative reconstructions of control and β-cat$^{ex3}$ iDCX neurons expressing RFP at 13 dpT. Scale bar = 20 μm.

F Quantification of dendritic length ($P$ < 0.0001), branch points ($P$ = 0.0495), and Sholl analysis ($P$ < 0.0001) showed a lower dendritic complexity of β-cat$^{ex3}$ iDCX neurons at 13 dpT (control: $n$ = 20 cells from four animals, β-cat$^{ex3}$ iDCX: $n$ = 19 cells from five animals).

Data information: Data represented as mean ± SEM, significance was determined using two-way ANOVA for Sholl analysis and two-tailed Mann–Whitney $U$-test for all other analyses, and significance levels are displayed in GP style (*$P$ < 0.0332, ***$P$ < 0.0002 and ****$P$ < 0.0001).

is highly dependent on the timing and dosage of β-catenin signaling activity.

## Age-dependent decrease of canonical Wnt signaling activity in the adult neurogenic lineage

Aging is accompanied by reduced Wnt production, activation of GSK3β, and increased expression of Wnt signaling inhibitors in the hippocampus (Okamoto *et al*, 2011; Miranda *et al*, 2012; Seib *et al*, 2013; Bayod *et al*, 2015). To determine whether these age-associated alterations translate into decreased canonical Wnt signaling in the DG, we analyzed BATGAL mice at different ages. In young adult, 8-week-old, BATGAL mice 49% of cells in the granule cell layer expressed the reporter. Previous studies have shown that adult neurogenesis in the murine DG is already severely compromised around the age of 5–6 months (Ben Abdallah *et al*, 2010; Trinchero *et al*, 2017). Notably, the percentage of reporter-positive cells was reduced to 26% in 24-week-old BATGAL mice, while in 36-week-old BATGAL mice only 21% of cells in the granule cell layer showed reporter activity (Fig 5A).

Next, we investigated whether the age-associated decrease in canonical Wnt signaling activity also affected the neurogenic lineage. Adult-born cells in young adult (i.e., 8-week-old) and middle-aged (i.e., 24-week-old) BATGAL mice were birthdated with BrdU. In young adult mice, the fraction of BrdU$^+$ cells expressing the immature marker DCX was high (82%) in 14-day-old cells and dropped to 12% in 28-day-old cells. Forty-two days after the BrdU pulse virtually all BrdU$^+$ cells had lost DCX expression, implying that all cells had reached a mature phenotype 42 days after birth (Fig 5B and C). Middle-aged mice featured a similar fraction of BrdU$^+$ cells expressing DCX at the 14-day-old time point; a higher percentage of cells, however, remained in the DCX$^+$ stage at 28 days (58%) and 42 days (15%). The delayed exit from the DCX$^+$ stage in middle-aged mice was paralleled by lower activity of canonical Wnt signaling. Fourteen days post-birthdating, the percentage of β-galactosidase-positive neurons among the BrdU-labeled cells was highly comparable between young and older mice (8-week-old: 14%, 24-week-old: 14%). While this percentage was increased in young adult mice at 28 dpi (37%), middle-aged mice exhibited only a negligible increase in β-galactosidase expression (19%). At 42 dpi, 52% of BrdU-labeled cells in young mice expressed β-galactosidase, while in older mice only 26% had active canonical Wnt signaling (Fig 5B and D). Collectively, these observations show that aging is associated with a

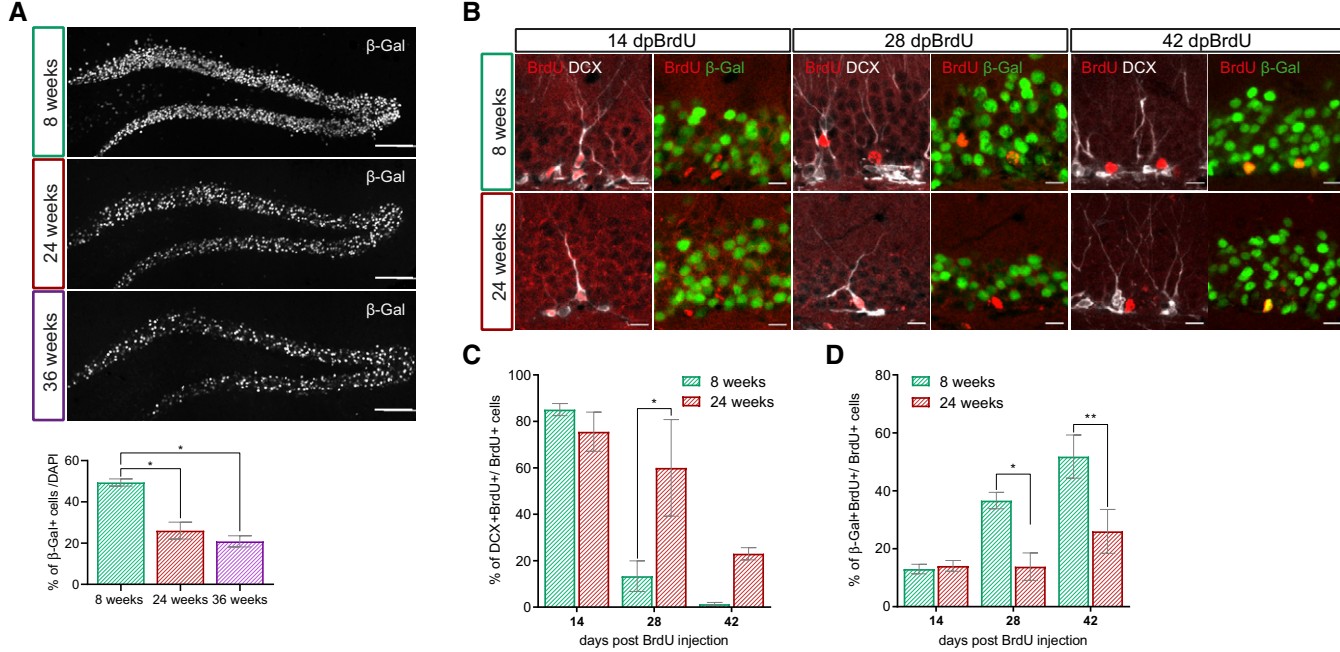

**Figure 5. Canonical Wnt signaling activity is decreasing with increasing age.**

A  Representative overview images and quantification of the β-galactosidase reporter signal in the DG of 8-week-old, 24-week-old, and 36-week-old BATGAL mice. Note the decrease in reporter signal with increasing age (*n* = 4 animals per age, 200–300 cells per animal; 8 weeks vs. 24 weeks *P* = 0.0268, 8 weeks vs. 36 weeks *P* = 0.0268). Scale bar = 100 μm.

B  Representative images of BrdU$^+$ cells (red) expressing DCX (gray) or β-galactosidase (green) at 14 days post-BrdU injection (dpBrdU), 28 and 42 dpBrdu in 8-week-old and 24-week-old BATGAL mice. Scale bar = 10 μm.

C  Percentage of BrdU$^+$ cells expressing DCX in 8-week-old and 24-week-old BATGAL mice (14 dpi: *n* = 3 animals, 28 dpi: *n* = 3 animals, 42 dpi: *n* = 3 animals). Note the prolonged expression of DCX in 24-week-old mice (28 dpBrdU *P* = 0.0155).

D  Percentage of BrdU$^+$ cells expressing the β-galactosidase reporter in 8-week-old and 24-week-old BATGAL mice (14 dpi: *n* = 3 animals, 28 dpi: *n* = 3 animals, 42 dpi: *n* = 3 animals). Note the reduced reporter expression in 24-week-old mice (28 dpBrdU *P* = 0.0117, 42 dpBrdU *P* = 0.0034).

Data information: Data represented as mean ± SEM, significance was determined using two-way ANOVA followed by Sidak's multiple comparison analysis, and significance levels are displayed in GP style (\**P* < 0.0332 and \*\**P* < 0.0021).

substantial decrease of canonical Wnt signaling in the adult neurogenic lineage.

## Enhanced β-catenin signaling activity in immature neurons of middle-aged mice accelerates maturation

Aging protracts dendritic growth and spine development of adult-born neurons (Beckervordersandforth *et al*, 2017; Trinchero *et al*,

2017). To determine whether increasing β-catenin signaling ameliorates age-associated maturation defects, we induced recombination in middle-aged (i.e., 24-week-old) β-cat[ex3] iDCX and control mice (Fig 6A). Three days after recombination, neurons in 24-week-old control mice displayed in comparison to neurons in 8-week-old control mice a much simpler dendritic arbor (Fig EV4A). Thirteen days after recombination, age-associated differences in dendritic arbor morphology were no longer present (Fig EV4A). Neurons in

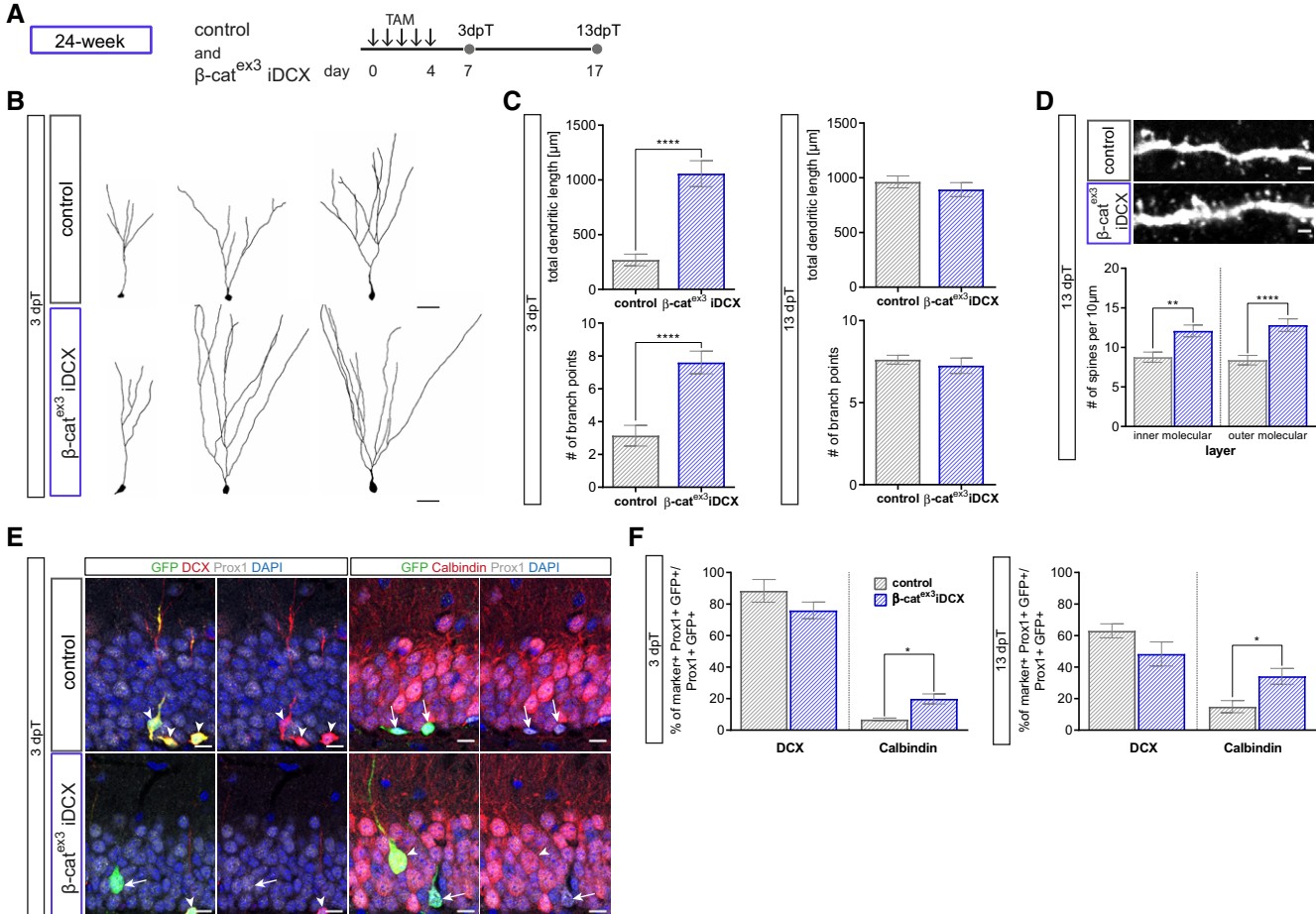

**Figure 6. Precocious activation of canonical Wnt signaling in middle-aged mice rescues aging phenotype.**

A  Experimental scheme to analyze the impact of β-catenin signaling activity on adult-born neuron development in middle-aged mice. Tamoxifen (TAM) was applied i.p. every 12 h for 5 days. Animals were analyzed 3 and 13 days post-tamoxifen (dpT).

B  Representative reconstructions of control and β-cat[ex3] iDCX neurons in 24-week-old mice at 3 dpT. Scale bar = 20 μm.

C  Dendritic length ($P < 0.0001$) and number of branch points ($P < 0.0001$) were increased in 24-week-old β-cat[ex3] iDCX compared to 24-week-old controls at 3 dpT (control: $n = 21$ cells from five animals, β-cat[ex3] iDCX: $n = 20$ cells from six animals). At 13 dpT, dendritic length ($P = 0.5788$) and number of branch points ($P = 0.4978$) were comparable (control: $n = 21$ cells from five animals, β-cat[ex3] iDCX: $n = 20$ cells from eight animals).

D  Representative images showing dendritic segment of the inner molecular layer of the DG in 24-week-old mice at 13 dpT. Quantification of the number of spines showed increased spine density in inner molecular and outer molecular layer in 24-week-old β-cat[ex3] iDCX mice at 13 dpT (iML $P = 0.0047$, oML $P < 0.0001$; control $n = 19$ cells from four animals, β-catex3 iDCX: $n = 20$ cells from four animals). Scale bars = 1 μm.

E  Representative images of recombined (GFP[+], green) neurons in 24-week-old animals at 13 dpT co-expressing Prox1 (gray) as a marker for neuronal fate and the stage-specific markers DCX (red) for immature neurons and Calbindin (red) for mature neurons. Arrows and arrowheads indicate marker-negative and marker-positive cells, respectively. Scale bar = 10 μm.

F  Quantification of marker expression. The fraction of Calbindin expressing cells was increased in the recombined population in β-cat[ex3] iDCX at 3 dpT ($P = 0.0081$) and 13 dpT ($P = 0.0259$; 3 dpT: control: $n = 4$ animals, β-cat[ex3] iDCX: $n = 9$ animals, 13 dpT: control: $n = 8$ animals, β-cat[ex3] iDCX: $n = 10$ animals).

Data information: Data represented as mean ± SEM, significance was determined using two-way ANOVA for Sholl analysis and two-tailed Mann–Whitney U-test for all other analyses, and significance levels were displayed in GP style (*$P < 0.0332$, **$P < 0.0021$ and ****$P < 0.0001$).

older mice, however, showed reduced expression of the mature neuron marker Calbindin (Fig EV4B) and substantially reduced spine densities (Fig EV4C). These data support the notion that increasing age is associated with a delay in maturation of adult-born neurons (Trinchero *et al*, 2017).

Intriguingly, recombined neurons in 24-week-old β-cat[ex3] iDCX mice featured a more mature dendritic arbor with longer dendrites and higher number of branch points compared to control neurons already 3 days after recombination (Fig 6B and C). Thirteen days after recombination, dendrite morphology was comparable between experimental groups (Fig 6C). Spine densities in β-cat[ex3] iDCX mice, however, were higher than in middle-aged control animals (Fig 6D) and were comparable to spine densities in young adult control mice (Fig EV4D). In addition, increased β-catenin activity resulted in a shift toward the expression of a mature marker profile. Three days after recombination, β-cat[ex3] iDCX neurons showed a trend toward lower DCX levels (control: 88%; β-cat[ex3] iDCX: 76%) and a small but significant increase in the expression of Calbindin (control: 7%; β-cat[ex3] iDCX: 20%; Fig 6E and F). Thirteen days after recombination, this shift toward a more mature marker profile was even more pronounced with 48% of neurons expressing DCX and 34% expressing Calbindin in β-cat[ex3] iDCX animals (control: DCX+ 63%; Calbindin+ 15%; Fig 6F). Notably, the proportion of mature Calbindin+ neurons in middle-aged β-cat[ex3] iDCX animals was highly similar to the proportion of Calbindin+ neurons in young adult mice (Fig EV4E).

To further substantiate the acceleration of maturation in middle-aged β-cat[ex3] iDCX animals, we determined the developmental trajectory of neurons of the same birthdate. Birthdating with BrdU or retrovirus was performed at the age of 24 weeks. Recombination was induced from day 10 until day 14 after the birthdating procedure (Fig 7A). Recombined cells were identified by GFP expression. Three days after recombination, β-cat[ex3] iDCX animals showed a trend toward a higher fraction of Calbindin+ neurons among BrdU-labeled neurons (Fig 7C). Thirteen days after recombination, the switch toward the expression of a mature marker profile became obvious: β-cat[ex3] iDCX animals showed a significant downregulation of DCX (β-cat[ex3] iDCX: 28%; control: 64%) and a substantial upregulation of Calbindin (β-cat[ex3] iDCX: 65%; control: 28%) in recombined neurons (Fig 7B and C). Notably, the fraction of Calbindin expressing neurons in 24-week-old β-cat[ex3] iDCX reached levels that were comparable to levels in young adult control mice (Fig EV4F). Three days after recombination, CAG-RFP birthdated neurons with activated β-catenin-dependent signaling displayed a substantially more mature dendrite morphology compared to control neurons (Fig 7D and E). Thirteen days after recombination, neurons in both experimental groups featured a dendritic arbor that was highly comparable to the dendritic arbor of neurons of the same birthdate generated in young adult mice (Figs 7F and G, and EV4G).

Collectively, these data demonstrate that increasing β-catenin signaling activity counters the age-associated protracted maturation of adult-born dentate granule neurons.

## Discussion

Our study identifies activity of canonical Wnt/β-catenin signaling as a regulator of neuronal maturation in adult hippocampal neurogenesis. The present data indicate that timing and dosage of β-catenin signaling regulate the tempo of dendritogenesis and the establishment of the highly stereotypic dendritic arbor of dentate granule neurons. Moreover, our data uncovers impaired canonical Wnt signaling activity as a target to alleviate the age-associated maturation defect of adult-born neurons.

Wnt-dependent signaling pathways serve pleiotropic functions in embryonic and adult neurogenesis. While canonical Wnt signaling has been primarily linked to early neurodevelopmental processes such as precursor proliferation and fate determination (Chenn & Walsh, 2002; Hirabayashi *et al*, 2004; Lie *et al*, 2005; Woodhead *et al*, 2006; Kuwabara *et al*, 2009), non-canonical Wnt signaling pathways are considered the main drivers of neural circuit formation and plasticity (Salinas, 2012; He *et al*, 2018; McLeod & Salinas, 2018). We demonstrate that dendritogenesis of adult-born dentate granule neurons is dependent on β-catenin signaling, which complements recent evidence that canonical Wnt/β-catenin signaling contributes to neuronal maturation and circuit formation in embryonic neurodevelopment (Martin *et al*, 2018; Ramos-Fernandez *et al*, 2019; Viale *et al*, 2019).

We found that in adult neurogenesis, genetic inhibition, and age-associated decrease of Wnt/β-catenin signaling activity were accompanied by a morphologically immature dendritic arbor and delayed dendritic development, respectively. In contrast, enhanced β-catenin activity by induction of the β-cat[ex3] transgene countered the age-associated delay in dendrite development, mature neuronal marker expression, and spine formation in middle-aged mice. These data strongly suggest that activity of the Wnt/β-catenin signaling pathway modulates the tempo of dendrite development and neuronal maturation and that the age-associated retardation of adult-born neuron maturation is caused by dampened Wnt/β-catenin signaling activity.

The observation that β-catenin signaling serves as a key regulator of dendrite growth and spine formation in adult hippocampal neurogenesis is surprising given the substantial evidence that Wnts regulate dendrite growth and spine formation of hippocampal neurons via local CamKII and JNK signaling (Rosso *et al*, 2005; Ciani *et al*, 2011; Ferrari *et al*, 2018). While it is possible that adult-born neuron development is regulated by highly distinct mechanisms, we would like to point out that our findings do not exclude that Wnt-induced CamKII and JNK signaling contribute to the regulation of dendrite growth and spine formation and co-operate with β-catenin signaling to regulate dentate granule neuron development.

We cannot exclude that the β-cat[ex3]-driven rescue of adult-born neuron maturation in middle-aged mice was at least in part caused by non-transcriptional functions of β-catenin (Yu & Malenka, 2003). The observation that inhibition of β-catenin signaling-driven transcriptional activity via dnLEF resulted in an immature neuronal morphology together with the finding that stabilization of β-catenin increased canonical Wnt signaling reporter activity in adult-born neurons, however, argues that β-catenin driven transcription has a significant impact on dendritic growth.

Our finding that expression of dnLEF and increasing β-catenin activity specifically in adult-born neurons inhibit and promote their dendrite growth, respectively, strongly indicate that β-catenin signaling regulates dendrite morphogenesis in a cell-autonomous fashion. High Wnt/β-catenin signaling activity is also detected in a large number of mature dentate granule neurons. It will be

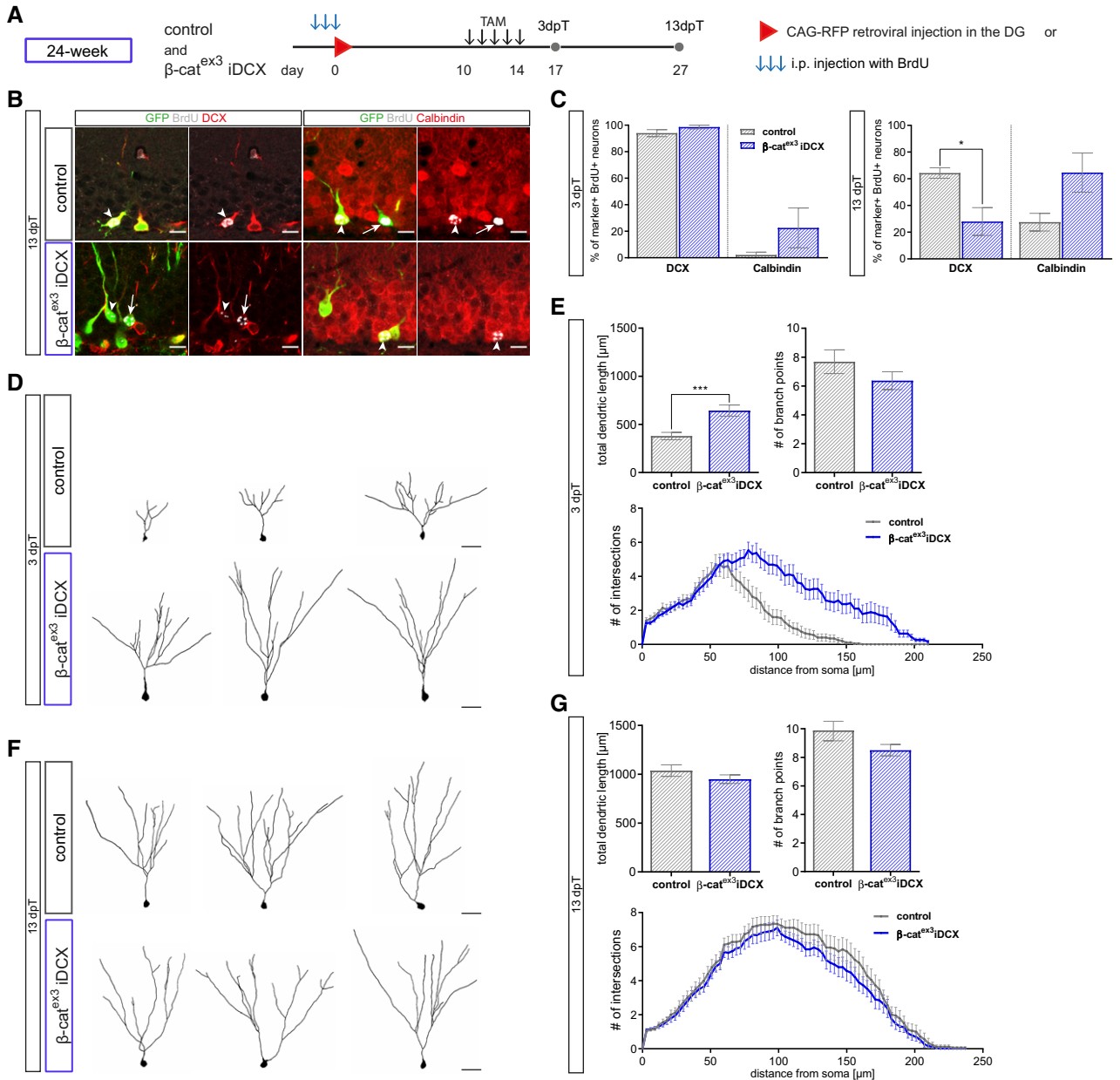

**Figure 7. Analysis of birthdated immature neurons in middle-aged mice confirms modulation of dendritic growth and spine formation by canonical Wnt signaling activity.**

A  Birthdating of adult-born neurons prior to recombination was conducted in 24-week-old mice. For marker expression analysis, BrdU was injected i.p. every 24 h for 3 days (B, C); for morphology analysis, CAG-RFP was stereotactically injected into the DG (D–G). Tamoxifen was applied i.p. every 12 h for 5 days. Mice were sacrificed 3 and 13 dpT.

B  Representative images of recombined (GFP[+], green) BrdU[+] (gray) neurons in 24-week-old control and β-cat[ex3] iDCX animals at 13 dpT expressing the stage-specific markers DCX (red) for immature neurons and Calbindin (red) for mature neurons. Arrows and arrowheads indicate marker-negative and marker-positive cells, respectively. Scale bar = 10 μm.

C  Quantification of DCX and Calbindin expression in BrdU[+] recombined cells. At 13 dpT, labeled neurons in β-cat[ex3] iDCX animals show a shift toward a more mature marker profile (DCX $P = 0.0238$, Calbindin $P = 0.0714$) (3 dpT: control: $n = 4$ animals, β-cat[ex3] iDCX: $n = 8$ animals; 13 dpT control: $n = 5$ animals, β-cat[ex3] iDCX: $n = 5$ animals).

D  Representative reconstructions of RFP-birthdated control and β-cat[ex3] iDCX neurons in 24-week-old mice at 3 dpT. Scale bar = 20 μm.

E  Quantification of dendritic length ($P = 0.0007$), branch points ($P = 0.3909$), and Sholl analysis ($P = 0.0001$) showed increased complexity of birthdated β-cat[ex3] iDCX cells at 3 dpT (control: $n = 20$ cells from four animals, β-cat[ex3] iDCX: $n = 19$ cells from five animals).

F  Representative reconstructions of RFP-birthdated control and β-cat[ex3] iDCX neurons in 24-week-old mice at 13 dpT. Scale bar = 20 μm.

G  Quantification of dendritic length ($P = 0.1857$), branch points ($P = 0.1331$), and Sholl analysis ($P = 0.1501$) showed no difference between β-cat[ex3] iDCX and control cells at 13 dpT (control: $n = 18$ cells from four animals, β-cat[ex3] iDCX: $n = 20$ cells from four animals)

Data information: Data represented as mean ± SEM, significance was determined using two-way ANOVA for Sholl analysis and two-tailed Mann–Whitney $U$-test for all other analyses, and significance levels are displayed in GP style (*$P < 0.0332$ and ***$P < 0.0002$).

interesting to determine, whether this activity plays a role in learning-induced dendrite growth of mature adult-born neurons (Lemaire *et al*, 2012) and increased spine formation of dentate granule neurons (O'Malley *et al*, 2000). Because newly generated neurons are highly dependent on synaptic input for survival (Tashiro *et al*, 2006a) and compete with mature neurons for synaptic input (Toni *et al*, 2007; McAvoy *et al*, 2016), modulation of dendrite growth and spine formation in mature neurons by Wnt/β-catenin signaling may also impact on the development and survival of newly generated neurons. Conversely, it would also be interesting whether enhanced β-catenin signaling and the resulting increase in spine formation endow newly generated neurons with an advantage during competition for synaptic input.

How β-catenin signaling regulates dendritogenesis and spine formation remains to be determined. Neurotrophin 3 was recently suggested to mediate dendritic growth and synapse formation downstream of β-catenin signaling during embryonic neurogenesis (Viale *et al*, 2019). Given the ability of neurotrophins to modulate dendritogenesis in adult hippocampal neurogenesis (Bergami *et al*, 2008; Chan *et al*, 2008; Trinchero *et al*, 2017), it is tempting to speculate that a similar β-catenin signaling/neurotrophin axis also operates during dendritogenesis of adult-born neurons.

In contrast to middle-aged mice, in which enhanced β-catenin activity rejuvenated the developmental trajectory and produced neurons with a normal dendritic morphology and physiological spine densities, increased β-catenin activity in young mice transiently accelerated dendritic growth but ultimately resulted in a stunted dendritic arbor and increased spine densities. We speculate that the differences in outcome between young adult and middle-aged mice are caused by age-associated differences in the tone of Wnt/β-catenin signaling in the neurogenic lineage. Middle-aged mice have reduced Wnt/β-catenin signaling in maturing neurons and induction of the β-cat$^{ex3}$ transgene may restore pathway activity to levels found in young adult mice. In contrast, expression of the β-cat$^{ex3}$ transgene in maturing neurons of young adult mice may result in excessive β-catenin signaling activity, which disrupts physiological dendritogenesis and spine development, which would be in line with the observation that restriction of β-catenin activity via a primary cilia-dependent mechanism is essential for dendritic refinement and correct synaptic integration of adult-born neurons (Kumamoto *et al*, 2012).

A key finding of this study is the demonstration that an ON-OFF-ON pattern of Wnt/β-catenin signaling activity parallels neurogenesis in the adult DG and that not only the reactivation of canonical Wnt signaling but also its initial attenuation are essential for dendritogenesis. Why canonical Wnt signaling activity has to be downregulated during early phases of neurogenesis remains to be determined. Given the evidence for crosstalk and functional antagonism between different Wnt pathways (Baksh *et al*, 2007; Mentink *et al*, 2018), we speculate that sustained β-catenin signaling interferes with non-canonical Wnt signaling pathways and their essential function in dendrite patterning and growth (Schafer *et al*, 2015; Goncalves *et al*, 2016a; Arredondo *et al*, 2019).

The mechanism underlying the reactivation of Wnt/β-catenin signaling remains to be determined. Downregulation of canonical signaling components was shown to drive the attenuation of Wnt/β-catenin signaling activity in the early neurogenic lineage (Schafer *et al*, 2015). Preliminary analysis of a published single-cell RNA

sequencing data set of the DG (Hochgerner *et al*, 2018) suggests a moderate increase in expression of genes associated with destabilization (APC, APC2, GSK3β, Cacybp) and inhibition of β-catenin (Ctnnbip1) in neuroblasts and immature neurons compared to precursors and mature neurons (Appendix Fig S1), which may contribute to the transient attenuation of Wnt/β-catenin signaling activity and explain its reactivation in maturing neurons. Another contributing factor may be that the adult-born neuron encounters new environments during its development. While the cell body remains in the DG and is continuously exposed to the same set of signals, the dendritic and axonal compartments gain access to potential new sources of Wnt ligands during their growth, such as the molecular layer, the hilus, and the CA3 region, which may trigger an increase in Wnt/β-catenin signaling activity. Interestingly, a previous report (Gogolla *et al*, 2009) and our analysis of an *in situ* hybridization database (http://www.brain-map.org) suggest that the CA3 region expresses Wnt ligands and modulators of canonical Wnt signaling (Appendix Fig S1).

Aging, neurodegenerative and neuropsychiatric diseases impede on the functional integration of adult-born hippocampal neurons (Li *et al*, 2009; Sun *et al*, 2009; Winner *et al*, 2011; Kim *et al*, 2012; Fitzsimons *et al*, 2013; Llorens-Martin *et al*, 2015; Trinchero *et al*, 2017). Considering the powerful modulation of dendrite and spine development by β-catenin signaling activity, the question arises whether aberrant Wnt/β-catenin signaling activity contributes to these pathologies and can be targeted to improve neurogenesis-dependent hippocampal plasticity. Our finding that the age-associated protraction of adult-born neuron maturation is paralleled by a substantial drop in canonical Wnt signaling activity and that activation of β-catenin signaling rejuvenated the time line of maturation-associated marker expression, dendrite growth and spine formation in middle-aged mice, renders Wnt/β-catenin signaling a promising candidate to ameliorate age-related impairment of hippocampal function.

# Material and Methods

### Experimental model and subject details

All experiments were carried out in accordance with the European Communities Council Directive (86/609/EEC) and were approved by the governments of Upper Bavaria and Middle-Franconia. For all experiments, mice were grouped housed in standard cages with *ad libitum* access to food and water under a 12 h light/dark cycle. BATGAL mice (Maretto *et al*, 2003) and Axin2$^{LacZ}$ mice (Lustig *et al*, 2002) have been described previously. To generate DCX:: CreER$^{T2}$; CAG-CAT-GFP; Ctnnb1$^{(ex3)fl/WT}$ animals, DCX::CreER$^{T2}$ mice (generated on a C57Bl6/J background (Zhang *et al*, 2010)), CAG-CAT-GFP mice (generated on a C57Bl6/J background (Nakamura *et al*, 2006)), and Ctnnb1$^{(ex3)fl}$ mice (generated on a C57Bl6/N background (Harada *et al*, 1999)) were crossed. DCX::CreER$^{T2}$; CAG-CAT-GFP; Ctnnb1$^{(ex3)fl/WT}$ were bred for > 10 generations. Subsequently, DCX::CreER$^{T2}$; CAG-CAT-GFP; Ctnnb1$^{(ex3)fl}$ and DCX::CreERT2; CAG-CAT-GFP; Ctnnb1$^{(ex3)wt}$ animals that were generated from the same cross were maintained as separate lines to enable homozygous breeding. DnLEF experiments were performed on mice with a mixed C57Bl6/J and C57Bl6/N background. Male and female mice were used for experiments.

## Method details

### Tissue processing

For brain tissue collection, mice were anesthetized using $CO_2$ and transcardially perfused with phosphate-buffered saline (PBS, pH 7.4) for 5 min at a rate of flow of 20 ml/min followed by fixation with 4% paraformaldehyde (PFA) in 0.1 mM phosphate buffer (pH 7.4, Roth, Cat# 0335) for 5 min at a rate of 20 ml/min. Brains were post-fixed in 4% PFA at 4°C overnight and subsequently dehydrated in 30% sucrose solution. Frozen brains were either coronally or sagittally cut using a sliding microtome (Leica Microsystems, Wetzlar, Germany). Sections were stored at −20°C in 96-well plates, filled with 200 µl cryoprotection solution per well.

### Genotyping

The following primers were used for genotyping DCX::CreER[T2], CAG-CAT-GFP, Ctnnb1[(ex3)fl] and BATGAL mice:

| DCX::CreER[T2]: | fwd | GTT TCA CTG GTT ATG CGG CG |
|---|---|---|
| | rev | GAG TTG CTT CAA AAA TCC CTT CC |
| CAT-eGFP: | fwd | ATT CCT TTG GAA ATC AAC AAA ACT |
| | rev | TGC TTT GAT ACT ATT CCA CAA ACC C |
| Ctnnb1[(ex3)fl]: | fwd | CTT CTC TGT GGG AAT AAA TGT TTG G |
| | rev | CTA CTT CAA GGA CAA GGG TGA CAG |
| BATGAL: | fwd | CGG TGA TGG TGC TGC GTT GGA |
| | rev | ACC ACC GCA CGA TAG AGA TTC |

### Tamoxifen administration

Tamoxifen (Sigma, Cat# T5648) was dissolved at a concentration of 10 mg/ml in ethanol and sunflower seed oil under constant agitation at room temperature. To induce recombination, 8-week-old and 24-week-old animals were intraperitoneally (i.p.) injected with 1 mg Tamoxifen every 12 h for five consecutive days (Mori *et al*, 2006).

### BrdU administration

Bromodeoxyuridine (BrdU, Sigma-Aldrich, Cat# B5002) was dissolved in sterile 0.9% NaCl solution at a concentration of 10 mg/ml. For birthdating experiments, 8-week-old, 24-week-old, and 36-week-old animals were intraperitoneally injected with 100 mg/kg BrdU three times: (i) every 2 h for the 30-min time point (described as 0 dpi in figures) (ii) every 24 h for all other time points.

### Histology and counting procedures

Immunofluorescence stainings were performed on 40- and 80-µm-thick free-floating brain slices. Selected brain slices were washed five times for 10 min with Tris-buffered saline (TBS) in netwell inserts at room temperature (RT). Blocking and permeabilization were conducted in blocking solution (3% donkey serum and 0.25% Triton X-100 in TBS) for 1 h at RT followed by incubation with the primary antibodies in blocking solution at 4°C for 72 h.

After rinsing five times in TBS for 10 min at RT, brain slices were incubated with secondary antibodies in blocking solution at 4°C overnight. After rinsing three times for 10 min in TBS at RT sections were incubated with 4′,6-Diamidino-2-phenylindole (Dapi, Sigma, Cat# D9542) for 10 min at RT and washed once for 10 min in TBS.

Brain slices were mounted on slides and coverslipped with Aqua Poly/Mount (Polysciences, Cat# 18606). Object slides were stored at 4°C in the dark.

If BrdU-incorporation into the DNA was to be examined, additional pretreatment in 2 N HCl was performed for 10 min at 37°C after staining for the other used antibodies was completed and fixated with 4% PFA for 10 min at RT. The slices were then incubated in 0.1 M Borate buffer for 10 min at RT and rinsed three times in TBS. Fluorescent staining for BrdU antibody was then performed as described above.

Primary antibodies were visualized with Alexa-conjugated secondary antibodies (all 1:1,000; Invitrogen). To amplify the GFP reporter signal, biotinylated secondary antibody (1:500; Vector Laboratories) was incubated at 4°C overnight. After rinsing the sections five times for 10 min with TBS at RT, Fluorophore-conjugated streptavidin (Invitrogen) in blocking solution was incubated overnight at 4°C and the staining procedure was finished as described above.

Confocal single plane images and z-stacks were taken with a Zeiss LSM 780 confocal microscope (Carl Zeiss, Oberkochen, Germany) equipped with four laser lines (405, 488, 559 and 633 nm) and 25×, 40×, and 63× objective lenses. As a standard, the number of total pixels per image and color depth was set to $1,024 \times 1,024$ and 16bit, respectively. For co-expression analysis, the 25× oil immersion objective was used; morphology was analyzed using the 40× and 63× oil immersion objective. Z-stack step size for co-expression and morphology analyses was set to 1.5 and 0.3 µm, respectively. Images were processed using Fiji ImageJ. 3D reconstructions were obtained using Imaris software (Bitplane AG, Zürich, Switzerland).

### Primary antibodies

| Antigen | Host | Manufacturer | RRID | Dilution |
|---|---|---|---|---|
| BrdU | Rat | Serotec | AB_609566 | 500 |
| Calbindin | Mouse | Swant | AB_10000347 | 250 |
| DCX | Goat | Santa Cruz Biosciences | AB_2088494 | 500 |
| DCX | Guinea pig | Merck Millipore | AB_1586992 | 1,000 |
| GFP | Chicken | Aves | AB_10000240 | 1,000 |
| Nestin | Mouse | Merck Millipore | AB_94911 | 500 |
| Prox1 | Rabbit | Merck Millipore | AB_177485 | 500 |
| RFP | Rat | Chromotek | AB_2336064 | 1,000 |
| Tbr2 | Rabbit | Abcam | AB_778267 | 500 |
| β-Galactosidase | Goat | Bio-Rad | AB_2307350 | 500 |
| β-Galactosidase | Chicken | Acris Antibodies GmbH | AB_11147602 | 500 |

### Retrovirus preparation and stereotactic injections

The retroviral plasmids CAG-GFP-IRES-Cre, CAG-dnLEF-IRES-GFP, and the CAG-RFP have been described (Tashiro *et al*, 2006a; Karalay *et al*, 2011; Steib *et al*, 2014). Production of replication

*Jana Heppt et al*

*The EMBO Journal*

incompetent MML-retroviruses was performed in human 293-derived retroviral packaging cell line (293GPG) 1F8 cells as described previously (Tashiro *et al*, 2006b).

For stereotactic injections, 8-week-old and 24-week-old mice under running conditions were deeply anesthetized by injecting 300 μl of a mixture of Fentanyl (0.05 mg/kg; Janssen-Cilag AG, New Brunswick, USA), Midazolam (5 mg/kg; Dormicum, Hoffmann-La Roche, Basel, Switzerland), and Medetomidine (0.5 mg/kg; Domitor, Pfizer Inc., New York, USA) dissolved in 0.9% NaCl. Mice were fixed in a stereotactic chamber, and the scalp was opened. Holes were drilled into the skull for injections into both hemispheres (coordinates from bregma were −1.9 anterior/posterior, ± 1.6 medial/lateral, −1.9 dorsal/ventral from dura). Nine hundred nanoliter virus particle suspension diluted with PBS to a concentration of $2 \times 10^8$ colony forming units/μl was injected into the DG of each hemisphere at a speed of 250 nl/min using a Digital Lab Standard Stereotaxic Instrument. Anesthesia was antagonized after surgery by injecting a mixture of Buprenorphine (0.1 mg/kg, Temgesic, Essex Pharma GmbH, Munich, Germany), Atipamezole (2.5 mg/kg, Antisedan, Pfizer Inc., New York, USA), and Flumazenil (0.5 mg/kg; Anexate, Hexal AG, Holzkirchen, Germany) dissolved in 0.9% NaCl.

For double injections with CAG-RFP and CAG-dnLEF-IRES-GFP, a virus particle suspension of both viruses with a concentration of $2 \times 10^8$ colony forming units/μl each was used. For survival analysis via co-injections of two viruses (Tashiro *et al*, 2006a; Jagasia *et al*, 2009), a cohort of animals was injected with the same virus particle suspension (CAG-RFP/CAG-dnLEF-IRES-GFP or CAG-GFP-IRES-Cre) and animals were sacrificed at 17 and 42 dpi.

## Quantification and statistical analysis

### Expression analysis of stage-specific markers

Co-expression analysis was conducted using ImageJ software. For cell stage marker (Nestin, Tbr2, DCX, Prox1, Calbindin) co-expression analysis, > 100 cells per animal were analyzed from at least four different animals. Expression of BrdU in BATGAL and $Axin2^{LacZ/+}$ mice was determined in at least two sections containing the hippocampus of at least four different animals and for control and β-cat$^{ex3}$ iDCX mice one section containing the hippocampus of at least five different animals. The number of biological replicates ($n$) analyzed is specified in the figure legends.

### Fluorescence intensity measurements

Fluorescence intensity of the β-galactosidase reporter as a proxy for canonical Wnt signaling activity was measured as corrected total cell fluorescence (CTCF) using ImageJ software. To quantify nuclear β-galactosidase expression level, an outline was drawn around the nucleus of a randomly chosen marker-positive cell a single in-focus plane and integrated density, area and mean fluorescence was measured, along with several adjacent background readings. Integrated density of the nuclear β-galactosidase signal was corrected by the area of the measured cell and the mean fluorescence of five background measurements of each staining (CTCF = integrated density − [area of selected cell × mean fluorescence of background readings]). CTCF values were visualized as dot blots with mean ± SEM.

### Dendritic and spine morphology analyses

To analyze detailed cell morphology, confocal images were taken using Zeiss LSM 780 confocal microscope (Carl Zeiss, Oberkochen, Germany) with a 40× and 63× oil immersion objective. For spine analysis, the digital zoom was set to three enabling better spatial resolution. Brain slices for dendritic morphology analysis and spine analysis were 80–100 and 40 μm thick, respectively. 3D reconstructions of neurons were obtained with Imaris using the Filament Tracer tool and the Surface Tracer tool. Values for total dendritic length, number of Sholl intersections, number of branch points, and basal dendrites were exported. The number of spines was investigated using Fiji.

### Survival analysis

Survival of neurons with dnLEF expression was determined using the retrovirus co-injection strategy described in Tashiro *et al* (2006a) and Jagasia *et al* (2009). Mice were stereotactically injected with a virus particle suspension of CAG-dnLEF-IRES-GFP and CAG-RFP retroviruses with a concentration of $2 \times 10^8$ colony forming units/μl each. The number of neurons expressing GFP or RFP was quantified in three different slices of four animals at 17 and 42 dpi and the ratio of GFP$^+$/RFP$^+$ was compared between both time points. For survival analysis of neurons with stabilized β-cat expression, β-cat$^{ex3}$ (i.e., Ctnnb1$^{(ex3)fl}$) mice and the respective β-cat$^{WT}$ (i.e., Ctnnb1$^{(ex3)WT}$) mice were stereotactically injected with the same virus particle suspension of CAG-GFP-IRES-Cre with a concentration of $2 \times 10^8$ colony forming units/μl. At 42 dpi, the number of transduced GFP$^+$ neurons was quantified in both cohorts.

### Statistical analysis

GraphPad Prism was used for statistical analysis. The statistical significance level α was set to 0.05. Gaussian distribution was tested using the Anderson–Darling test, D'Agostino Pearson omnibus test, Shapiro–Wilk test, and Kolmogorov–Smirnov test. If not applicable, non-Gaussian distribution was assumed. Statistical significance was determined using the two-tailed Mann–Whitney *U*-test, Kruskal–Wallis test followed by Dunn's test for multiple comparisons and two-way ANOVA followed by Sidak multiple comparisons test for Sholl analysis and significance levels were displayed in GP style ($*P < 0.0332$, $**P < 0.0021$ and $***P < 0.0002$, $****P < 0.0001$). Unless otherwise stated in the figure legend, results are represented as mean ± SEM. The number of biological replicates ($n$) is specified for each analysis in the figure legend. For marker expression experiments, $n$ equals the number of individual animals analyzed and for morphology analysis $n$ equals the number of neurons analyzed from a minimum of three different animals.

# Data availability

This study includes no data deposited in external repositories.

**Expanded View** for this article is available online.

## Acknowledgements

We thank S. Jessberger and all members of the Lie laboratory for helpful discussions and comments on the manuscript. This work was supported

by grants from the German Research Foundation (LI 858/6-3 and 9-1 to D.C.L, INST 410/45-1 FUGG), the Bavarian Research Network "ForIPS" and "ForINTER" to D.C.L., the University Hospital Erlangen (IZKF grants E12, E16, E21 to D.C.L.). J.H. and M.T.W. are members of the research training group 2162 "Neurodevelopment and Vulnerability of the Central Nervous System" funded by the Deutsche Forschungsgemeinschaft (270949263/DFG GRK2162/1). Open access funding enabled and organized by Projekt DEAL.

## Author contributions

Conceptualization: JH, NP, DCL; Investigation: JH, M-TW, IS, CB, JZ, DV-W; Formal analysis: JH, M-TW, JZ, DCL; Resources and funding acquisition: WW, DCL; Reagents: MMT, WW; Writing—original draft, JH, DCL; Writing—review and editing: JH, DCL; Supervision: NP, DCL.

## Conflict of interest

The authors declare that they have no conflict of interest.

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
