## [Review Process File · The EMBO Journal]

β -catenin signaling modulates the tempo of dendritic growth of adult-born hippocampal neurons

Jana Heppt, Marie-Theres Wittman, Iris Schaeffner, Charlotte Billmann, Jingzhong Zhang, Daniela Vogt-Weisenhorn, Nilima Prakash, Wolfgang Wurst, Makoto Mark Taketo, and Chichung Lie
DOI: [10.15252/embj.2020104472](https://doi.org/10.15252/embj.2020104472)

Corresponding authors: Chichung Lie (chi.lie@fau.de)

Review Timeline:

Submission Date:	14th Jan 20
Editorial Decision:	13th Mar 20
Revision Received:	26th Jun 20
Editorial Decision:	3rd Aug 20
Revision Received:	6th Aug 20
Accepted:	11th Aug 20

Editor: Karin Dumstrei

Transaction Report:

Dear Dieter,

Thank you for submitting your manuscript to the EMBO Journal. I am sorry for the delay in getting back to you with a decision, but I have now received the comments from the three referees.

As you can see below, the referees appreciate that the analysis adds new insight and find the study the interesting. They raise a number of good points that I would like to invite you to address in a revised version.

I should add that it is EMBO Journal policy to allow only a single major round of revision, and that it is therefore important to resolve the major concerns at this stage. acceptance of your manuscript will therefore depend on the completeness of your responses in this revised version.

I am happy to discuss the raised points further and maybe it would be most helpful to do so via phone or skype. I will contact you in the few days to discuss this further.

Thank you for the opportunity to consider your work for publication. I look forward to your revision.

with best wishes

Karin

Karin Dumstrei, PhD
Senior Editor
The EMBO Journal

- a point-by-point response to the referees' comments, with a detailed description of the changes made (as a word file).

- a word file of the manuscript text.

- individual production quality figure files (one file per figure)

- a complete author checklist, which you can download from our author guidelines (<https://www.embopress.org/page/journal/14602075/authorguide>).

- Expanded View files (replacing Supplementary Information)

Further information is available in our Guide For Authors:

The revision must be submitted online within 90 days; please click on the link below to submit the revision online before 11th Jun 2020.

Link Not Available

Referee #1:

In this manuscript, "Canonical Wnt-signaling modulates the tempo of dendritic growth of adult-born hippocampal neurons" by Heppt et al., the authors extensively investigate canonical Wnt signal activation and its relation to neuronal differentiation in the adult hippocampal neuronal differentiation, suggesting that the biphasic activity pattern of Wnt signal is required for proper dendrite development. In addition, they showed that enhancement of Wnt signal activation restored the deficit in neuronal maturation caused by age associated decline of Wnt signal activation. Their experimental designs are solid and nicely performed with convincing data to support their conclusion, however, there are a few points which the author should address prior to be accepted.

Specific comments

1. As for Figure 1B, it is very hard to judge the b-gal expression level with merged pictures. They should provide b-gal single staining picture to show the expression in each cell of interest.

2. Although they calculated % of (marker+ g-cal+ cell)/(marker+ cell) in Figure 1C, this is based on whether the cells are positive or negative. Therefore, the data does not provide the activation level like the authors described, i.e., low, moderate and high. It would be great if they measure the sum (or average) of signal intensity of each cell of interest to indicate the activation level. And once they confirm that the values are correlated with those of % of (marker+ g-cal+ cell)/(marker+ cell), I think they can use % of (marker+ g-cal+ cell)/(marker+ cell) for the quantification of the activation as they presented in the manuscript afterwards.

3. I wonder how canonical Wnt signal activation level is biphasic in cells in neuronal lineage. Is this caused by change of expression of Wnt receptor or its ligand in this region? I suggest the authors examining the expression of these proteins (or mRNAs) in the cells of interest. Immunohistochemistry, in situ hybridization, RT-PCR of isolated marker-positive cells....., only one of these experiments is enough.

Referee #2:

This manuscript by Heppt and colleagues examines the role of canonical Wnt signalling in dendritic development of adult-born neurons in the hippocampus. Using a combination of transgenic reporter lines and loss of function of Wnt signalling the authors report a biphasic pattern of activation of canonical Wnt signalling that regulates dendrite growth in the neurogenic niche. They also present data on the impact of ageing in the response to activation of canonical Wnt signalling.

The study is of a great interest in the field and the authors made a number of important observations. However, the paper is not clearly written and there are parts that are difficult to follow. The authors claimed that they addressed the role of Wnt signalling by performing loss and gain of function studies. However, they only performed loss of function experiments by expressing a dominant negative LEF in figure 1. All the other studies are based on the expression of an activated β -catenin. The authors also concluded at the end of the introduction and in the discussion that their "data reveal a new cell autonomous function of the canonical Wnt signalling ...". This conclusion is incorrect. The authors mainly tested the role of β -catenin in specific cells of the neurogenic niche in the hippocampus. But this does not imply that the canonical Wnt pathway works in a cell-autonomous manner. In addition, the authors did not examine the impact of modulating β -catenin on neighbouring cells (where β -catenin is not activated) to reach this conclusion. In addition, there is a concern about the variability in the morphology of control neurons presented making the conclusions difficult. The authors did not discuss previous findings that demonstrate that non-canonical Wnt signaling regulates dendritogenesis in the hippocampus. Although these studies were not focused on the neurogenic niche, the authors should discuss these apparent surprising results. In its current form, this paper is not acceptable for publication in Embo J.

Specific comments:

- 1)Figure 1: the different level of Wnt activation in different cell types is not clear in the images. A change in the choice of the color for the Nestin, Tbr2 or Calbindin could help the visualisation. The authors should use green rather than grey.
- 2)Figure 1B and C, there is no quantification. Fig 1D: there is no statistics. Why?
- 3)It would be better to present the graphs with SEM rather than SD.

- 4) The morphology of control neurons is not consistent. While it is understandable that neurons in older animals will have shorter and less complex dendrites, control neurons from similar ages should have comparable morphology. For example between Fig 2H and Fig 3B. Why is the morphology so different? In addition, the authors should be consistent how the morphology depicted. Some neurons have cell bodies whereas others don't.
- 5) It would be important to determine the impact of loss of function (dnLEF and activated β -catenin) on the rate of cell death in the neurogenic niche.
- 6) Some of the effects are very small. For example, Figure 3G on the % of DCX and Calbindin cells is relatively small. What would the biological impact be with such small difference?
- 7) The observation of the impact of ageing in the number of BATGAL cells was really interesting. However, the legend of Figure 4 B-D was not clear that the parameters measured were done in BATGAL cells. This should be mentioned in the figure legend. The authors should also present at least one image showing *lax*⁺ cells, DCX and BrdU.
- 8) Figure 5H. Why is the staining for the different markers so different from those presented in previous figures?

Minor comments:

Many references in the text have the name of two others where others followed the normal citation (the last name of the first author and et), This needs to be corrected.
Some of the references are incomplete (missing volume and pages).

Referee #3:

In the present study, Heppt J. et al. addressed the question whether beyond its well-established role in adult hippocampal neural stem cells, canonical Wnt-signaling levels also temporally tune dendritic arbor formation of adult-born hippocampal neurons. Towards this, the authors used a variety of genetic tools in order to silence or activate canonical Wnt signaling in newborn neurons during different time windows of the maturation process, and performed a detailed morphological analysis of these manipulated neurons. They provide evidence for a biphasic activity of canonical Wnt-signaling with high levels of Wnt signaling in stem cells and more mature stages of neurogenesis, well separated by a drop in Wnt signaling during immature stages. This drop and subsequent raise appear to play a critical role in the appropriate development of those neurons' dendrites. Moreover, canonical Wnt signaling declines with aging and the authors show that counteracting this reduction in Wnt/ β -catenin signaling, already in middle-aged mice, restores some of the deficits in dendritic growth and spine formation. The conclusions are based on well-designed experiments and complement previous knowledge on non-canonical Wnt signaling in dendritogenesis. However, there are a few issues that should be addressed:

1. What is the experimental evidence for the dominant negative effect of dnLEF on Wnt signaling?
2. How do the levels of induced Wnt signaling compare to physiological levels?
3. In all quantifications throughout the manuscript, the individual data points (rather than average + sem) and real p-values should be shown.
4. The strength of the morphological analyses would benefit if the number of analyzed neurons was increased. This applies in particular to Fig 2 (only 11 control cells were analyzed in panels D-E) and

Fig 3 (only 13 cells per group for spine density measurement in panel E; 12 cells in the gain-of-function group in panel M).

5. Fig.3 and Fig.5 contain some redundant data obtained using different genetic models of enhanced Wnt signaling with distinct experimental precision. The experiments using the DCX promoter to drive CreERT2 suffer from the fact that newborn neurons express DCX during a relatively broad time window, and hence cohorts of adult-born hippocampal neurons of different maturation stages will be targeted. Results from these experiments are therefore only partially conclusive. The authors then addressed the same issue using a temporally much better defined approach of birth-dating, which allowed targeting of specific cohorts of DCX+ cells. Consequently, the effects appear to be more pronounced. For this reason, I would suggest to relegate the information of the less clear models to the supplementary material.

6. However, there would be an appealing rationale to use the DCX driven CreERT2 model if the authors addressed potential behavioral consequences (similar to e.g. McAvoy et al. Neuron 2016) of improving dendritogenesis in aged animals.

Minor points:

1. In the main text related to Fig. 1D, it is stated that newborn neurons were birth-dated by a single pulse of BrdU. This is not consistent with schematic representation in the figure and figure legend, which indicate 3 injections were made.

2. McAvoy et al (Neuron 2016) have shown that there is competition between cohorts of newly generated and mature granule neurons. It would be interesting to learn whether any of the manipulations altering the dendritic arborisation and spine formation also affects earlier born mature neurons in the same direction or whether enhanced dendritogenesis occurs at expense of dendrites and spines in mature neurons. While this may be out of scope of this study, it would be worth to be discussed. Along these lines, does canonical Wnt signaling regulate the expression of Klf9?

3. How many cells in how many animals were analysed to obtain the percentages in the bottom paragraph of page 8?

4. Correct the incomplete sentence on page 11 starting after ";" whereas...

Response to reviewers' comments

Referee #1:

In this manuscript, "Canonical Wnt-signaling modulates the tempo of dendritic growth of adult-born hippocampal neurons" by Heppt et al., the authors extensively investigate canonical Wnt signal activation and its relation to neuronal differentiation in the adult hippocampal neuronal differentiation, suggesting that the biphasic activity pattern of Wnt signal is required for proper dendrite development. In addition, they showed that enhancement of Wnt signal activation restored the deficit in neuronal maturation caused by age associated decline of Wnt signal activation. Their experimental designs are solid and nicely performed with convincing data to support their conclusion, however, there are a few points which the author should address prior to be accepted.

Answer: We are glad to learn that the reviewer considers the data convincing.

Specific comments

Comment 1: As for Figure 1B, it is very hard to judge the b-gal expression level with merged pictures. They should provide b-gal single staining picture to show the expression in each cell of interest.

Answer: Thank you for this suggestion. We have added a magnification of single cells expressing β -Galactosidase and the specific stage marker and have adjusted the color schemes of the pictures (as suggested by reviewer 2).

Comment 2: Although they calculated % of (marker+ g-cal+ cell)/(marker+ cell) in Figure 1C, this is based on whether the cells are positive or negative. Therefore, the data does not provide the activation level like the authors described, i.e., low, moderate and high. It would be great if they measure the sum (or average) of signal intensity of each cell of interest to indicate the activation level. And once they confirm that the values are correlated with those of % of (marker+ g-cal+ cell)/(marker+ cell), I think they can use % of (marker+ g-cal+ cell)/(marker+ cell) for the quantification of the activity as they presented in the manuscript afterwards.

Answer: Thank you for this suggestion. The reviewer correctly states that reporter intensity per cell may provide a better indication of the activation level than the number of reporter positive cells. We have now measured the fluorescence intensity of the reporter signal in stage-specific marker positive cells and BrdU-birthdated cells as a proxy for the activation level. This analysis supports the notion that canonical Wnt-signaling follows a biphasic activity pattern in the adult neurogenic lineage. The findings are documented in (Fig. 1D and G). We also added measurements of the fluorescence intensity of β -Galactosidase in BATGAL mice upon transduction of CAG-dnLEF-IRES-GFP and in β -cat^{ex3} iDCX BATGAL mice to validate inhibition and activation of β -catenin signaling (Reviewer 3's comment 1+2). The findings are documented in (Fig. EV1 and EV3).

Comment 3: I wonder how canonical Wnt signal activation level is biphasic in cells in neuronal lineage. Is this caused by change of expression of Wnt receptor or its ligand in this region? I suggest the authors examining the expression of these proteins (or mRNAs) in the cells of interest. Immunohistochemistry, in situ hybridization, RT-PCR of isolated marker-positive cells....., only one of these experiments is enough.

Answer: We agree with the reviewer that this is one of the important questions that should be addressed in the future. As suggested by the reviewer we screened for differentially expressed genes with an annotation for Wnt signaling according to the Kyoto Encyclopedia of Genes and Genomes (KEGG, mmu 04310). To determine the expression along dentate granule neuron maturation we analyzed a single cell RNA-sequencing data set from the Linnarson laboratory (Hochgerner et al., 2018). An overview of the genes with stage specific expression was added to the extended view section (Fig. EV5). While expression of Wnt receptors, β -catenin and β -catenin associated transcription factors remained constant during lineage progression, genes associated with β -catenin destabilization (APC, APC2, GSK3 β , Cacybp) and inhibition (Ctnnbip1) appeared to be moderately increased in neuroblasts and immature neurons compared to precursors and mature neurons (Fig. EV5A), which may contribute to the transient attenuation of Wnt/ β -catenin signaling activity and its re-activation in maturing neurons. Furthermore, we found by screening the Allen mouse brain atlas (<http://www.brain-map.org>) that mRNA for canonical Wnt-signaling ligands (Wnt7b and Rspo2) are expressed in the CA3 region (Fig. EV5B), which may result in enhanced canonical Wnt-signaling once the growing axon reaches its target region. These preliminary data are now included in the extended view section and described in the text as follows:

“...The mechanism underlying the reactivation of Wnt/ β -catenin signaling remains to be determined. Down-regulation of canonical signaling components was shown to drive the attenuation of Wnt/ β -catenin signaling activity in the early neurogenic lineage (Schafer et al., 2015). Preliminary analysis of a published single cell RNA sequencing data set of the dentate gyrus (Hochgerner et al., 2018) suggests a moderate increase in expression of genes associated with destabilization (APC, APC2, GSK3 β , Cacybp) and inhibition of β -catenin (Ctnnbip1) in neuroblasts and immature neurons compared to precursors and mature neurons (Fig. EV5A), which may contribute to the transient attenuation of Wnt/ β -catenin signaling activity and explain its re-activation in maturing neurons. Another contributing factor may be that the adult-born neuron encounters new environments during its development. While the cell body remains in the dentate gyrus and is continuously exposed to the same set of signals, the dendritic and axonal compartments gain access to potential new sources of Wnt-ligands during their growth, such as the molecular layer, the hilus and the CA3 region, which may trigger an increase in Wnt/ β -catenin signaling activity. Interestingly, a previous report (Gogolla et al., 2009) and our analysis of an in-situ hybridization data base (<http://www.brain-map.org>) suggests that the CA3 region expresses Wnt-ligands and modulators of canonical Wnt-signaling (Fig. EV5B).”

Referee #2:

This manuscript by Heppt and colleagues examines the role of canonical Wnt signalling in dendritic development of adult-born neurons in the hippocampus. Using a combination of transgenic reporter lines and loss of function of Wnt signalling the authors report a biphasic pattern of activation of canonical Wnt signalling that regulates dendrite growth in the neurogenic niche. They also present data on the impact of ageing in the response to activation of canonical Wnt signalling. The study is of a great interest in the field and the authors made a number of important observations.

Answer: We are glad to learn that the reviewer considers our findings interesting for the field.

General comment: However, the paper is not clearly written and there are parts that are difficult to follow. The authors claimed that they addressed the role of Wnt signalling by performing loss and gain of function studies. However, they only performed loss of function experiments by expressing a dominant negative LEF in figure 1. All the other studies are based on the expression of an activated β -catenin. The authors also concluded at the end of the introduction and in the discussion that their "data reveal a new cell autonomous function of the canonical Wnt signalling ...". This conclusion is incorrect. The authors mainly tested the role of β -catenin in specific cells of the neurogenic niche in the hippocampus. But this does not imply that the canonical Wnt pathway works in a cell-autonomous manner. In addition, the authors did not examine the impact of modulating β -catenin on neighbouring cells (where β -catenin is not activated) to reach this conclusion.

Answer:

Canonical Wnt-signaling signals through stabilization of β -catenin. BATGAL reporter mice, dnLEF and stabilized β -catenin are commonly used tools to interrogate the function of canonical Wnt/ β -catenin signaling in different systems. In the revised manuscript we have now added the validation of these tools for their use in studying adult hippocampal neurogenesis: 1) We injected the dnLEF encoding retrovirus and a control retrovirus into the dentate gyrus of BATGAL mice. We found that reporter expression is significantly reduced in dnLEF transduced neurons. 2) We crossed the BATGAL reporter into β -cat^{ex3} iDCX and control iDCX mice and compared reporter activity in recombined cells. Here, we found a significant increase in recombined neurons in β -cat^{ex3} iDCX. These validations have been included in the revised manuscript (Figs. EV1 and EV3).

We, however, agree with the reviewer that while being a common tool to interrogate the function of canonical Wnt/ β -catenin signaling, stabilized β -catenin in the strictest sense interrogates β -catenin mediated signaling. We have therefore changed the wording in the sections that describe experiments employing stabilized β -catenin and use the phrase β -catenin signaling instead of canonical Wnt-signaling.

We also agree with the reviewer that while the experiments using dnLEF and stabilized β -catenin show an effect in cells expressing the transgene (i.e., cell-autonomous effect), we cannot exclude the possibility that β -catenin signaling in neighboring cells may also have an effect on the maturation of adult-born neurons. We are discussing this possibility in the revised manuscript. The respective section now reads as follows:

"...High Wnt/ β -catenin signaling activity is also detected in a large number of mature dentate granule neurons. It will be interesting to determine, whether this activity plays a role in learning induced dendrite growth of mature adult-born neurons (Lemaire et al, 2012) and increased spine formation of dentate granule neurons (O'Malley et al, 2000). Because newly generated neurons are highly dependent on synaptic input for survival (Tashiro et al, 2006a) and compete with mature neurons for synaptic input (McAvoy et al, 2016; Toni et al, 2007), modulation of dendrite growth and spine formation in mature neurons by Wnt/ β -catenin signaling may also impact on the development and survival of newly generated neurons. Conversely, it would also be interesting whether enhanced β -catenin-signaling and the resulting increase in spine formation endow newly generated neurons with an advantage during competition for synaptic input..."

General comment (continued): In addition, there is a concern about the variability in the morphology of control neurons presented making the conclusions difficult.

Answer: This point has also been raised in the specific comment section. Please see our answer to Specific Comment 4 below.

General comment (continued): The authors did not discuss previous findings that demonstrate that non-canonical Wnt signaling regulates dendritogenesis in the hippocampus. Although these studies were not focused on the neurogenic niche, the authors should discuss these apparent surprising results.

Answer: Thank you for this comment. In our initial manuscript we had described the notion that non-canonical Wnt signaling drives neural circuit formation and plasticity. We agree that it would be interesting to specifically discuss previous findings on non-canonical Wnt-signaling in dendritogenesis and spine formation in hippocampal neurons. The discussion has been extended and now reads as follows:

“...The observation that β -catenin signaling serves as a key regulator of dendrite growth and spine formation in adult hippocampal neurogenesis is surprising given the substantial evidence that Wnts regulate dendrite growth and spine formation of hippocampal neurons via local CamKII and JNK signaling (Ciani et al, 2011; Ferrari et al, 2018; Rosso et al, 2005). While it is possible that adult-born neuron development is regulated by highly distinct mechanisms, we would like to point out that our findings do not exclude that Wnt-induced CamKII and JNK signaling contribute to the regulation of dendrite growth and spine formation and co-operate with β -catenin signaling to regulate dentate granule neuron development...”

Specific comments:

Comment 1: Figure 1: the different level of Wnt activation in different cell types is not clear in the images. A change in the choice of the color for the Nestin, Tbr2 or Calbindin could help the visualisation. The authors should use green rather than grey.

Answer: We have adjusted the color schemes of the pictures and added a magnification of single cells expressing β -Galactosidase and the specific stage marker to improve visibility (also suggested by reviewer 1).

Comment 2: Figure 1B and C, there is no quantification. Fig 1D: there is no statistics. Why?

Answer: For better readability we did not describe the quantification of the reporter positive cells in the text but referred the reader to the graphs in Figure 1B and 1C. In the revised manuscript we are describing the values in the figure legend. The statistics for the data has been added.

Comment 3: It would be better to present the graphs with SEM rather than SD.

Answer: We have changed the graph presentation to mean \pm SEM.

Comment 4: The morphology of control neurons is not consistent. While it is understandable that neurons in older animals will have shorter and less complex dendrites, control neurons from similar ages should have comparable morphology. For example between Fig 2H and Fig 3B. Why is the morphology so different? In addition, the authors should be consistent how the morphology depicted. Some neurons have cell bodies whereas others don't.

Answer: We originally depicted cell bodies in Figure 2 for better visualization of the dendrites exciting the cell body horizontally or on the basal site. Following the suggestion of this reviewer, we are now presenting the cell bodies in all reconstructions.

The reviewer correctly notes, that the dendrite morphology of birth-dated control neurons varies between experiments. We suspect that these differences may be caused by the different mouse backgrounds in the individual experiments: *Ctnnb1*^{(ex3)^{fl}} were originally generated in a C57Bl6/N background (Harada et al., 1999). *DCX::CreERT2* mice (Zhang et al. 2010) and *CAG-CAT-GFP* mice (Kawamoto et al. 2000) were originally generated in a C57Bl6/J background. Differences between the C57Bl6/N and the C57Bl6/J substrain with regard to physiology and behavior have previously been noted (Ahlgren and Voitkar, 2019).

- In the experiments depicted in Figure 3 *DCX::CreERT2; CAG-CAT-GFP; Ctnnb1*^{(ex3)^{wt}} served as controls for *DCX::CreERT2; CAG-CAT-GFP; Ctnnb1*^{(ex3)^{fl}} mice. These lines were also analyzed to study the effects of aging and of enhanced β -catenin signaling.
- In the experiments depicted in Figure 2H we used *Ctnnb1*^{(ex3)^{fl}} and the respective *Ctnnb1*^{(ex3)^{WT}} mice (control).
- Experiments depicted in Figure 2B were conducted in mice of a mixed C57Bl6/J and C57Bl6/N background. These mice were co-injected with dnLEF encoding retrovirus and an RFP encoding retrovirus, the latter serving as an internal control.

We would argue that the conclusions are not affected by the differences in dendrite morphologies between controls in the individual experiments. We do understand the reviewer's comment and describe the different genetic backgrounds in the method section, which now reads as follows:

"...To generate DCX::CreERT2; CAG-CAT-GFP; Ctnnb1(ex3)fl/WT animals, DCX::CreERT2 mice [generated on a C57Bl6/J background (Zhang et al., 2010)], CAG-CAT-GFP mice [generated on a C57Bl6/J background (Nakamura et al., 2006)], and Ctnnb1(ex3)fl mice [generated on a C57Bl6/N background (Harada et al., 1999)], were crossed. DCX::CreERT2; CAG-CAT-GFP; Ctnnb1(ex3)fl/WT were bred for > 10 generations. Subsequently, DCX::CreERT2; CAG-CAT-GFP; Ctnnb1(ex3)fl and DCX::CreERT2; CAG-CAT-GFP; Ctnnb1(ex3)wt animals that were generated from the same cross were maintained as separate lines to enable homozygous breeding. DnLEF experiments were performed on mice with a mixed C57Bl6/J and C57Bl6/N background...."

Comment 5: It would be important to determine the impact of loss of function (dnLEF and activated β -catenin) on the rate of cell death in the neurogenic niche.

Answer: We agree that this is an important question and performed additional experiments to determine the survival of neurons in the context of expression of dnLEF and the expression of stabilized β -catenin. In the case of dnLEF expression we co-injected animals with a control retrovirus (CAG-RFP) and the dnLEF encoding CAG-dnLEF-IRES-GFP retrovirus and quantified the ratio of dnLEF transduced GFP+ neurons to RFP+ neurons expression at 17 days post injection (dpi) and 42dpi. We observed a substantial decrease in this ratio (GFP+/RFP+ cells) from approximately 1.5 (at 17 dpi) to 0.5 (at 42 dpi), indicating that a large fraction of neurons with dnLEF expression do not survive long-term. The impact of failure to attenuate β -catenin on survival was analyzed by injecting the identical amounts of the CAG-GFP-IRES-Cre retrovirus into the dentate gyrus of control and

β -cat^{ex3} mice. Analysis was performed 42 dpi. At this time-point control mice showed approximately three times more transduced neurons than β -cat^{ex3} mice, suggesting that enhancing β -catenin starting at the level of fast dividing precursor cells impairs long-term survival.

The new data are now included in Figure EV2 and described as follows:

“... Dendritic arborization in the molecular layer provides the structural basis for formation of glutamatergic synaptic input from the entorhinal cortex. Previous studies identified glutamatergic input as a critical signal for survival of adult-born neurons (Tashiro et al, 2006a). To determine the long-term survival of dnLEF-transduced neurons, mice were co-injected with CAG-dnLEF-IRES-GFP and CAG-RFP and analyzed at 17dpi and 42dpi (Fig. EV2A). At 42dpi, the number of dnLEF transduced neurons was dramatically reduced and the ratio of GFP+ to RFP+ cells dropped from approximately 1.5 at 17dpi to 0.5 at 42dpi, indicating that dnLEF expression strongly decreased survival of adult-born neurons. Moreover, dnLEF expressing neurons featured a dendritic morphology with subtle alterations in the Sholl analysis (Fig. EV2B-D) ...”

and

“... To determine how failure to attenuate β -catenin signaling affected the long-term fate of neurons, a second cohort of mice was analyzed at 42 days post viral injection (Fig. EV2E). The number of transduced neurons was dramatically reduced in β -catex3 mice, suggesting that continuous β -catenin dependent signaling impaired long-term survival of adult-born neurons. Moreover, Sholl analysis revealed subtle alterations in dendrite morphology of the remaining β -catex3 neurons (Fig. EV2F-H). ...”

Comment 6: Some of the effects are very small. For example, Figure 3G on the % of DCX and Calbindin cells is relatively small. What would the biological impact be with such small difference?

Answer: The marker expression in combination with dendrite morphology and spine density served as read-out for the degree of maturity. We agree with the reviewer that the effects of enhanced β -catenin expression on marker expression as documented in Figure 3G - while statistically significant – are relatively small. The effects of on dendrite growth (at the 3 day time-point after recombination) and on spine density (at the 13 day time-point) are, however, substantial.

In general, the physiological impact of the loss of DCX expression and the gain of Calbindin in adult-born neurons is unknown. Hence, we cannot at this point answer the question what the biological impact of a slight increase in Calbindin expression and loss of DCX expression would be. We, however, consider it important to report these small effects, given that under conditions of reduced β -catenin signaling (i.e., aging), there is a substantial effect of enhanced β -catenin signaling on the timing of the DCX to Calbindin switch.

Comment 7: The observation of the impact of ageing in the number of BATGAL cells was really interesting. However, the legend of Figure 4 B-D was not clear that the parameters measured were done in BATGAL cells. This should be mentioned in the figure legend. The authors should also present at least one image showing laz+ cells, DCX and BrdU.

Answer: Thank you for this comment. We have adapted the figure legend and are now describing that the analysis was done in BATGAL reporter mice. In addition, representative images showing β -Galactosidase, DCX and BrdU have been added (Fig. 4B).

Comment 8: Figure 5H. Why is the staining for the different markers so different from those presented in previous figures?

Answer: Some of the perceived differences may stem from the fact that we initially chose a different color scheme for Figure 5H and that the images do not include nuclear counterstaining with DAPI. A certain degree of difference in the staining/image is expected, as the staining procedure for BrdU requires a harsh pretreatment with HCl, which impacts on the quality of the staining. Moreover, HCl pretreatment is incompatible with DAPI counterstaining.

For consistency, we revised Figure 5H and adjusted the color scheme to match the color scheme of the other figures.

Minor comments:

Many references in the text have the name of two others where others followed the normal citation (the last name of the first author and et), This needs to be corrected.

Some of the references are incomplete (missing volume and pages).

Answer: We apologize for this mistake. The references have been corrected.

Referee #3:

In the present study, Heppt J. et al. addressed the question whether beyond its well-established role in adult hippocampal neural stem cells, canonical Wnt-signaling levels also temporally tune dendritic arbor formation of adult-born hippocampal neurons. Towards this, the authors used a variety of genetic tools in order to silence or activate canonical Wnt signaling in newborn neurons during different time windows of the maturation process, and performed a detailed morphological analysis of these manipulated neurons. They provide evidence for a biphasic activity of canonical Wnt-signaling with high levels of Wnt signaling in stem cells and more mature stages of neurogenesis, well separated by a drop in Wnt signaling during immature stages. This drop and subsequent raise appear to play a critical role in the appropriate development of those neurons' dendrites. Moreover, canonical Wnt signaling declines with aging and the authors show that counteracting this reduction in Wnt/ β -catenin signaling, already in middle-aged mice, restores some of the deficits in dendritic growth and spine formation. The conclusions are based on well-designed experiments and complement previous knowledge on non-canonical Wnt signaling in dendritogenesis. However, there are a few issues that should be addressed:

Answer: We appreciate that reviewer 3 recognizes our effort to demonstrate the role of canonical Wnt signaling in maturation of adult born hippocampal dentate granule neurons.

Comment 1: What is the experimental evidence for the dominant negative effect of dnLEF on Wnt signaling?

Answer: DnLEF is a truncated version of the transcription factor LEF that lacks the β -catenin binding domain but binds to the consensus sequence on the DNA acting as a repressor until replaced by full length LEF (van de Wetering et al., 1996; Hovanes et al., 2001). In the adult

hippocampus it was shown that expression of dnLEF was sufficient to suppress transcription of the Wnt-target gene Prox1 (Karalay et al., 2011), indicating that β -catenin dependent transcription of target genes is inhibited by dnLEF expression.

To validate the inhibitory effect of dnLEF expression on canonical Wnt signaling, we injected a MML retrovirus bi-cistronically encoding for dnLEF and GFP into the dentate gyrus of BATGAL mice. BATGAL mice injected with a retrovirus encoding for GFP served as control. The number of reporter positive cells and the corrected total cell fluorescence of transduced cells was quantified. The fraction of reporter positive cells as well as average β -Galactosidase expression levels were significantly reduced in dnLEF transduced neurons, which demonstrates that expression of dnLEF inhibited canonical Wnt signaling-induced transcriptional activity. This validation experiment has been included into the revised manuscript and is illustrated in Figure EV1.

Comment 2: How do the levels of induced Wnt signaling compare to physiological levels?

Answer: As per suggestion of reviewer 1 and following this reviewer's comment, we have determined the fluorescence signal of β -Galactosidase in BATGAL mice as a proxy for the level of Wnt/ β -catenin signaling. To validate the induction of β -catenin signaling in β -cat^{ex3} iDCX mice we crossed the BATGAL reporter mouse line with the β -cat^{ex3} iDCX and the control mouse line. Recombined neurons showed on average higher reporter expression levels than recombined neurons in control mice, indicating that tamoxifen-induced recombination increased β -catenin signaling activity in β -cat^{ex3} iDCX mice (Fig. EV3C,D). Notably, recombined cells are surrounded by non-recombined cells in the granule cell layer, which are most likely mature dentate granule neurons. Importantly, these surrounding, non-recombined cells clearly show higher reporter expression than recombined neurons (Fig. EV3C), suggesting that β -cat^{ex3} driven β -catenin signaling activity did not exceed physiological β -catenin signaling activity levels found in dentate granule neurons. These findings are described in the revised manuscript, which reads as follows:

"... We first validated the tamoxifen-mediated induction of β -catenin dependent transcription in β -cat^{ex3} iDCX. To this end, the β -cat^{ex3} iDCX and the control mouse line were crossed with the BATGAL reporter mouse line. Recombination was induced in 8-week old mice by injection of tamoxifen on five consecutive days. Animals were analyzed 13 days after the tamoxifen pulse (Fig. EV3B). GFP+ recombined cells in β -cat^{ex3} iDCX; BATGAL mice showed on average higher reporter expression levels than recombined cells in control mice, indicating that tamoxifen-induced recombination increased canonical Wnt signaling activity in β -cat^{ex3} iDCX mice (Fig. EV3C, D). Numerous non-recombined cells in the granule cell layer, which were most likely mature dentate granule neurons, showed higher reporter expression than recombined neurons (Fig. EV3C), suggesting that β -cat^{ex3} driven β -catenin signaling activity did not exceed physiological Wnt/ β -catenin signaling activity levels found in dentate granule neurons...."

Comment 3: In all quantifications throughout the manuscript, the individual data points (rather than average + sem) and real p-values should be shown.

Answer: We have received the opposing suggestion from reviewer 2 to present the graphs as mean + SEM and have followed his/her suggestion to display values such as dendrite length, branch points, marker expression and spine densities.

For quantification of the fluorescence activity we followed this reviewer's suggestion and display the data as a dot plot, as there is considerable variability of the individual values and information would be lost by only displaying the mean values.

The description of the p-value is now consistently provided in the legend.

Comment 4: The strength of the morphological analyses would benefit if the number of analyzed neurons was increased. This applies in particular to Fig 2 (only 11 control cells were analyzed in panels D-E) and Fig 3 (only 13 cells per group for spine density measurement in panel E; 12 cells in the gain-of-function group in panel M).

Answer: We agree with the reviewer and have increased the number of analyzed neurons.

Comment 5: Fig.3 and Fig.5 contain some redundant data obtained using different genetic models of enhanced Wnt signaling with distinct experimental precision. The experiments using the DCX promoter to drive CreERT2 suffer from the fact that newborn neurons express DCX during a relatively broad time window, and hence cohorts of adult-born hippocampal neurons of different maturation stages will be targeted. Results from these experiments are therefore only partially conclusive. The authors then addressed the same issue using a temporally much better defined approach of birth-dating, which allowed targeting of specific cohorts of DCX+ cells. Consequently, the effects appear to be more pronounced. For this reason, I would suggest to relegate the information of the less clear models to the supplementary material.

Answer: We understand the comment. In the course of this revision, we had prepared an alternative version and had moved the respective parts of figure 3 and of figure 5 to the extended version figures. We, however, received the comment from several independent readers that the restructuring of the figures significantly impeded the readability of the manuscript. For this reason we decided to keep the original structure of figure 3 and 5. Should the reviewer, however, feel that we should rearrange the figures, we would be happy to do so.

Comment 6: However, there would be an appealing rationale to use the DCX driven CreERT2 model if the authors addressed potential behavioral consequences (similar to e.g. McAvoy et al. Neuron 2016) of improving dendritogenesis in aged animals.

Answer: We fully agree that it would be interesting to study potential behavioral consequences of the manipulation of β -catenin signaling. We, however, believe that the complex behavioral analysis is beyond the scope of the present study and also cannot perform such time-consuming analyses within the limited timeframe allowed for revision.

Minor points:

Comment 1: In the main text related to Fig. 1D, it is stated that newborn neurons were birth-dated by a single pulse of BrdU. This is not consistent with schematic representation in the figure and figure legend, which indicate 3 injections were made.

Answer: We apologize for this error. The schematic representation and description in the figure legend is correct and we changed the respective sentence in the main text:

"... To further assess the time course of canonical Wnt signaling activity in adult neurogenesis, newborn cells in 8-week-old reporter mice were birthdated with Bromodeoxyuridine (BrdU)..."

Comment 2: McAvoy et al (Neuron 2016) have shown that there is competition between cohorts of newly generated and mature granule neurons. It would be interesting to learn whether any of the manipulations altering the dendritic arborisation and spine formation also affects earlier born mature neurons in the same direction or whether enhanced dendritogenesis occurs at expense of dendrites and spines in mature neurons. While this may be out of scope of this study, it would be worth to be discussed. Along these lines, does canonical Wnt signaling regulate the expression of Klf9?

Answer: Thank you for this interesting comment. We have extended the discussion to include this thought and to respond to the comment of reviewer 2, who rightly pointed out that modulation of β -catenin signaling in neighboring cells may also have an effect on maturation of adult-born neurons. The respective section of the discussion reads as follows:

“...High Wnt/ β -catenin signaling activity is also detected in a large number of mature dentate granule neurons. It will be interesting to determine, whether this activity plays a role in learning induced dendrite growth of mature adult-born neurons (Lemaire et al, 2012) and increased spine formation of dentate granule neurons (O'Malley et al, 2000). Because newly generated neurons are highly dependent on synaptic input for survival (Tashiro et al., 2006a) and compete with mature neurons for synaptic input (McAvoy et al, 2016; Toni et al, 2007), modulation of dendrite growth and spine formation in mature neurons by Wnt/ β -catenin signaling may also impact on the development and survival of newly generated neurons. Conversely, it would also be interesting whether enhanced β -catenin-signaling and the resulting increase in spine formation endow newly generated neurons with an advantage during competition for synaptic input...”

We have not tested whether canonical Wnt-signaling regulates the expression of the spine-destabilizing transcription factor Klf9 in adult-born neurons. Circumstantial evidence from other systems suggests that Klf9 may regulate the expression of genes associated with Wnt-signaling (Knödler et al., 2017; Pabona et al., 2012). While it will be interesting to investigate the potential relationship between Wnt-signaling and Klf9, we have decided not to include this point in the discussion, given the lack of data and supporting literature.

Comment 3: How many cells in how many animals were analysed to obtain the percentages in the bottom paragraph of page 8?

Answer: For analysis of the age-associated decrease in canonical Wnt-signaling one section from 4 different animals per time point was quantified. The chosen sections derived from a comparable area the hippocampus. Per section 200-300 cells were quantified resulting in approximately 1000 quantified cells per age. The information has been added to the figure legend.

Comment 4: Correct the incomplete sentence on page 11 starting after ";" whereas...

Answer: Thank you for the comment. We have corrected the sentence. The respective section now reads as follows:

“...We found that in adult neurogenesis, genetic inhibition and age-associated decrease of Wnt/ β -catenin-signaling activity were accompanied by a morphologically immature dendritic arbor and delayed dendritic development, respectively. In contrast enhanced β -catenin activity by induction of the β -cat^{ex3} transgene countered the age-associated delay in dendrite

development, mature neuronal marker expression and spine formation in middle-aged mice...

Dear Chichung,

Thanks for submitting your manuscript to The EMBO Journal. Your study has now been re-reviewed by the referees and their comments are provided below. As you can see the referees appreciate the added data and support publication. Referee #2 has one final text change suggestion. Given the input from the referees, I am therefore very pleased to accept the manuscript for publication here. Before sending you the final accept letter, we just need the following editorial points being addressed:

Could you take a look at the layout of Figure 3 and 5, I find it a bit difficult to navigate the figures. Please take a look. Also, Fig 3J and Fig 5J panels are missing

Please double check scale bars in 1B, EV1B

Fig EV5 spans multiple pages, which is not optimum. Could we maybe make it into an appendix figure? The appendix file opens as a PDF and so if the figure runs over multiple pages not a problem.

Is the data used for Fig 5N beta-cat 24 weeks used in EV4G as well? If so please mention this in the figure legend.

We also need a Data Availability section. As far as I can see no data is generated that needs to be deposited in a database and if so then please state: This study includes no data deposited in external repositories

I have asked our publisher to do their pre-publication checks on the paper. They will send me the file within the next few days. Please wait to upload the revised version until you have received their comments.

We include a synopsis of the paper (see <http://emboj.embopress.org/>). Please provide me with a general summary statement and 3-5 bullet points that capture the key findings of the paper.

We also need a summary figure for the synopsis. The size should be 550 wide by [200-400] high (pixels). You can also use something from the figures if that is easier.

That should be all let me know if you have any questions. Congratulations on a nice paper

With best wishes

Karin

Karin Dumstrei, PhD
Senior Editor
The EMBO Journal

- a point-by-point response to the referees' comments, with a detailed description of the changes made (as a word file).

- a word file of the manuscript text.

- individual production quality figure files (one file per figure)

- a complete author checklist, which you can download from our author guidelines (<https://www.embopress.org/page/journal/14602075/authorguide>).

- Expanded View files (replacing Supplementary Information)

Further information is available in our Guide For Authors:

The revision must be submitted online within 90 days; please click on the link below to submit the revision online before 1st Nov 2020.

Link Not Available

Referee #1:

As far as I am concerned, the authors addressed all points of criticisms raised by the reviewers. The manuscript now seems to meet the requirements to stand as a good article for this journal.

Referee #2:

The authors have carefully revised the manuscript and answered all the questions and suggestions made by the reviewers.

The authors provided a point by point answer to all the questions the three reviewers had.

I am satisfied with the authors' response.

However, I do have a further query. In page 5, first paragraph the authors wrote in the revised manuscript "Both Reporter lines showed a qualitatively comparable biphasic pattern of canonical Wnt signalling activity (Fig 1B, C)." It is not clear to me where the evidence for a biphasic pattern is. The same statement was present in the previous version but I forgot to question it when I sent my comments. The authors should clarify this statement.

Other than this query, I believe the manuscript is appropriate for publication in EMBO Journal.

Referee #3:

Major points were addressed satisfactorily.

The authors performed the requested editorial changes.

Response to the comment of Reviewer 2:

Comment 1:

However, I do have a further query. In page 5, first paragraph the authors wrote in the revised manuscript "Both Reporter lines showed a qualitatively comparable biphasic pattern of canonical Wnt signalling activity (Fig 1B, C)." It is not clear to me where the evidence for a biphasic pattern is. The same statement was present in the previous version but I forgot to question it when I sent my comments. The authors should clarify this statement.

Response: The evidence for the biphasic pattern is provided in Figure 1 B-G, which shows that canonical Wnt-signaling activity is high in neural stem / precursor cells, attenuated in immature neurons, and reactivated in mature neurons. As the statement of biphasic activity precedes the presentation of the data we have modified the sentence, which now reads as follows: "...Both reporter lines showed a qualitatively comparable pattern of canonical Wnt signaling activity (Fig. 1B, C)...."

Thank you for the very productive reviewing process and your advise.

Dear Chichung,

Thanks for submitting your revised manuscript to The EMBO Journal. I have now had a chance to take a careful look at everything and all looks good.

I am therefore very pleased to accept the manuscript for publication here. Congratulations on a nice study!

With best wishes

Karin

Karin Dumstrei, PhD
Senior Editor
The EMBO Journal

Please note that it is EMBO Journal policy for the transcript of the editorial process (containing referee reports and your response letter) to be published as an online supplement to each paper. If you do NOT want this, you will need to inform the Editorial Office via email immediately. More information is available here: http://emboj.embopress.org/about#Transparent_Process

Your manuscript will be processed for publication in the journal by EMBO Press. Manuscripts in the PDF and electronic editions of The EMBO Journal will be copy edited, and you will be provided with page proofs prior to publication. Please note that supplementary information is not included in the proofs.

Should you be planning a Press Release on your article, please get in contact with embojournal@wiley.com as early as possible, in order to coordinate publication and release dates.

If you have any questions, please do not hesitate to call or email the Editorial Office. Thank you for your contribution to The EMBO Journal.

Corresponding Author Name:

Journal Submitted to:

Manuscript Number: